# Dual vaccination against IL-4 and IL-13 protects against chronic allergic asthma in mice

Eva Conde[1,2,3,8], Romain Bertrand[3,8], Bianca Balbino[1,2], Jonathan Bonnefoy[3], Julien Stackowicz[1,2], Noémie Caillot[3], Fabien Colaone[3], Samir Hamdi[3], Raïssa Houmadi[4], Alexia Loste[4], Jasper B. J. Kamphuis [4], François Huetz [1], Laurent Guilleminault[4,5], Nicolas Gaudenzio[4], Aurélie Mougel[4], David Hardy[6], John N. Snouwaert[7], Beverly H. Koller[7], Vincent Serra[3], Pierre Bruhns[1,9], Géraldine Grouard-Vogel[3,9] & Laurent L. Reber [1,4,9✉]

Allergic asthma is characterized by elevated levels of IgE antibodies, type 2 cytokines such as interleukin-4 (IL-4) and IL-13, airway hyperresponsiveness (AHR), mucus hypersecretion and eosinophilia. Approved therapeutic monoclonal antibodies targeting IgE or IL-4/IL-13 reduce asthma symptoms but require costly lifelong administrations. Here, we develop conjugate vaccines against mouse IL-4 and IL-13, and demonstrate their prophylactic and therapeutic efficacy in reducing IgE levels, AHR, eosinophilia and mucus production in mouse models of asthma analyzed up to 15 weeks after initial vaccination. More importantly, we also test similar vaccines specific for human IL-4/IL-13 in mice expressing human IL-4/IL-13 and the related receptor, IL-4Rα, to find efficient neutralization of both cytokines and reduced IgE levels for at least 11 weeks post-vaccination. Our results imply that dual IL-4/IL-13 vaccination may represent a cost-effective, long-term therapeutic strategy for the treatment of allergic asthma as demonstrated in mouse models, although additional studies are warranted to assess its safety and feasibility.

[1] Unit of Antibodies in Therapy and Pathology, Institut Pasteur, UMR 1222 INSERM, Paris, France. [2] Sorbonne University, ED394, Paris, France. [3] Neovacs SA, Paris, France. [4] Toulouse Institute for Infectious and Inflammatory Diseases (Infinity), INSERM UMR1291, CNRS UMR5051, University Toulouse III, Toulouse, France. [5] Department of respiratory medicine, Toulouse University Hospital, Faculty of Medicine, Toulouse, France. [6] Institut Pasteur, Experimental Neuropathology Unit, Paris, France. [7] Department of Genetics, University of North Carolina at Chapel Hill, Chapel Hill, NC, USA. [8] These authors contributed equally: Eva Conde, Romain Bertrand. [9] These authors jointly supervised this work: Pierre Bruhns, Géraldine Grouard-Vogel, Laurent L. Reber. ✉email: laurent.reber@inserm.fr

Asthma is the most common chronic lung disease, affecting >300 million people worldwide, and with at least 250,000 deaths attributed to the disease each year[1]. An estimate of 20% of asthma patients suffer from uncontrolled, moderate-to-severe asthma[2], presenting with persistent symptoms, with reduced lung functions and recurrent exacerbations, despite the use of high-dose pharmacological therapy[3]. The heterogeneity of asthma phenotypes represents a challenge for adequate assessment and treatment of the disease[4]. However, type 2 inflammation characterized by production of interleukin-4 (IL-4) and IL-13 in the lung, airway eosinophilia, and high levels of IgE antibodies occurs in ~50% of patients with asthma[1,5].

Even though IL-4 and IL-13 present similar structures and share one receptor subunit (IL-4Rα)[6], IL-4 and IL-13 are also thought to have some nonredundant functions in allergy and asthma[7]. In particular, IL-4 is considered to act predominantly in the early phase of asthma development through its role in regulating T cell proliferation and survival, and IgE synthesis[6]. In contrast, IL-13 would predominantly be involved in late phases of allergic reactions, such as airway remodeling and mucus hypersecretion[6].

Phase 3 studies indicated that dupilumab—a monoclonal antibody (mAb) against IL-4Rα that blocks both IL-4 and IL-13 signaling[8]—is efficient at decreasing the rate of severe exacerbations, and at improving lung function in patients with moderate-to-severe asthma[9]. Dupilumab was approved in 2018 as an add-on maintenance treatment in moderate-to-severe asthma with type 2 inflammation. However, use of this (or any other) mAb in chronic asthma is limited by high cost and the need to perform injections over years to lifelong. Therefore, while IL-4 and IL-13 are now clinically validated therapeutic targets for the treatment of asthma, there is a clear need to improve current strategies, with the goal of reaching long-term cost-effective therapeutic effects.

Conjugate vaccines called kinoids can elicit an endogenous, long-lasting neutralizing antibody response against a given cytokine[10], and could be a favorable alternative to therapeutic mAb administration. Vaccination against mouse IL-4 partially reduced IgE levels and eosinophilia with minor effects on mucus hypersecretion in a mouse asthma model[11]. A recombinant mouse IL-13 peptide-based virus-like particle vaccine had significant effects on mucus production without, however, affecting IgE levels[12]. Based on these partial results, and on the superior clinical efficacy in human asthma of targeting both IL-4 and IL-13 signaling (i.e., dupilumab) rather than targeting either IL-4 or IL-13 alone[13–15], we hypothesized that a dual vaccination against IL-4 and IL-13 would be particularly potent at reducing the severity of chronic asthma.

Here, we design conjugate vaccines against IL-4 and IL-13 rather than IL-4Rα to minimize the risk that these vaccines may induce antibodies capable of activating this receptor or inducing antibody-dependent cellular toxicity. We show that prophylactic and therapeutic dual vaccination against mouse IL-4 and IL-13 reduces key features of chronic allergic asthma in mice. We also demonstrate the immunogenicity of a vaccine targeting human IL-4/IL-13 in a novel mouse strain humanized for IL-4, IL-13, and IL-4Rα. Overall, our results suggest that dual IL-4/IL-13 vaccination is a promising long-term therapeutic strategy for allergic asthma, pending further safety and feasibility assessment in additional preclinical models.

## Results

### Anti-mouse IL-4 and IL-13 kinoids induce potent and long-lasting neutralizing responses.

We developed mouse IL-4 and IL-13 kinoids (IL-4-K and IL-13-K), by coupling these cytokines with diphtheria "cross-reactive material 197" (CRM197, a nontoxic mutant of diphtheria toxin used as a carrier protein in a number of approved conjugate vaccines[16]) using a thiol-maleimide conjugation (Supplementary Figs. 1 and 2). Mice were immunized intramuscularly with IL-4-K and IL-13-K alone or in combination (or the carrier protein CRM197 alone as a control), combined 1:1 (v:v) with SWE, a squalene oil-in-water emulsion adjuvant[17] (Fig. 1a). We did not observe visible adverse effects of the vaccines, as mice had normal behavior, and vaccination with kinoids had no effect on body weight (Supplementary Fig. 3). Immunization with IL-4-K and/or IL-13-K induced high anti-IL-4 and/or anti-IL-13 antibody titers, respectively, detectable already 6 weeks after primary immunization (Supplementary Fig. 4a, b). As expected, all mice exposed to CRM197 or kinoids developed anti-CRM197 antibodies (Supplementary Fig. 4c). Importantly, anti-cytokine antibodies generated upon vaccination with the kinoids exhibited strong neutralizing capacities against the respective cytokine in >90% of mice starting 6 weeks after primary immunization (Fig. 1b, c). Such neutralizing capacity could still be detected in >60% of the mice over 1 year after primary immunization (Supplementary Fig. 5). Importantly, conjugation between cytokine and carrier protein was required for potent antibody responses and for neutralizing activity (Supplementary Fig. 6). Altogether, these data indicate that efficient long-term neutralization of both IL-4 and IL-13 can be achieved through vaccination with kinoids.

### Combined vaccination against IL-4 and IL-13 protects against chronic asthma in mice.

We then tested the prophylactic efficacy of these vaccines in a chronic asthma model. Mice received a total of 12 intranasal (i.n.) administrations of *Dermatophagoides farinae* house dust mite (HDM) extract (one of the major allergens in human asthma[18]) over a period of 6 weeks, a protocol known to reproduce key features of human chronic asthma[19] (Fig. 1a). We assessed airway responses in this asthma model using invasive plethysmography[20]. HDM-treated control mice exhibited marked increase in lung resistance and elastance upon exposure to aerosolized methacholine (a bronchoconstrictor used as part of the diagnostic of asthma in human[21]), as compared to PBS-treated control mice (Fig. 1d, e). These two features were significantly reduced in mice vaccinated with the IL-4-K, and absent in mice vaccinated with the IL-13-K or the combination of IL-4-K and IL-13-K (Fig. 1d, e). The superior benefit of the dual IL-4/IL-13 vaccine on airway hyperresponsiveness (AHR) was even more apparent when using noninvasive whole-body plethysmography[22] (Supplementary Fig. 7). Our data suggest that AHR may be more dependent on IL-13 than IL-4 in this chronic asthma model, and can be prevented upon prophylactic dual vaccination against IL-4 and IL-13.

Next, we assessed the effect of the kinoids on airway eosinophilia and inflammation. HDM-treated mice demonstrated elevated eosinophil numbers in bronchoalveolar lavage (BAL) fluid that were reduced fivefold and sixfold in mice vaccinated with IL-4-K or IL-13-K, respectively, and reduced even further (21-fold) upon dual vaccination (Fig. 1f). The benefit of the dual vaccination was even more evident when assessing eosinophil numbers that reduced eightfold in lung tissue, whereas single vaccination with IL-4-K or IL-13-K alone had no effect on eosinophil numbers (Fig. 1g). Consistent with these results, levels of IL-5—a type 2 cytokine which plays a key role in eosinophilic inflammation[5]—were also markedly reduced in BAL fluid of HDM-treated mice vaccinated with IL-4-K and/or IL-13-K (Fig. 1h). By contrast, we observed similar low levels of IL-4+ and IL-13+ CD4+ T cells (Supplementary Fig. 8a, b), as well as similar numbers of IL-4+ type 2 innate lymphoid cells (ILC2) (Supplementary Fig. 8c, d) in the lungs of HDM-treated controls

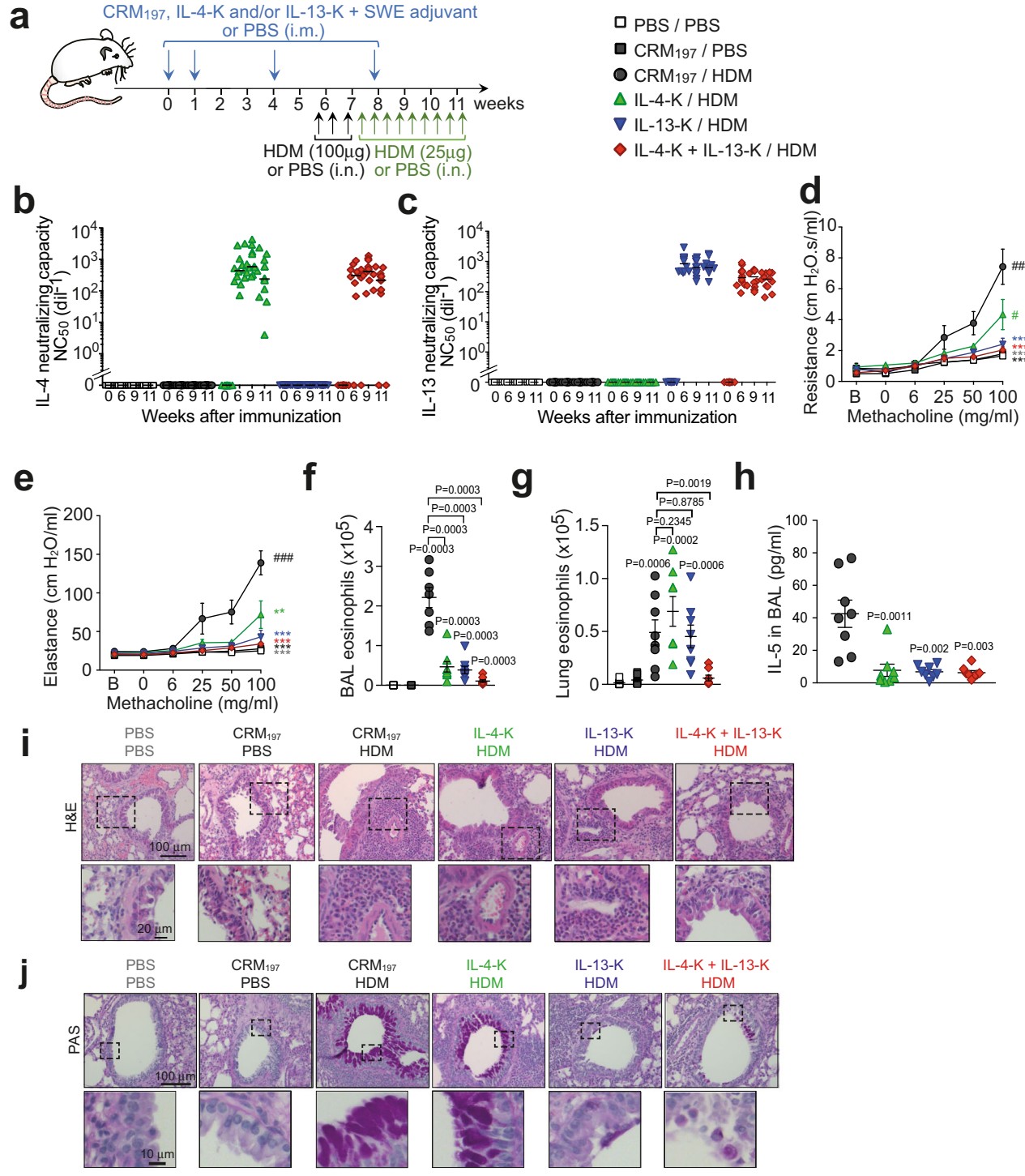

and vaccinated mice. Altogether, these results suggest that vaccination with IL-4-K or IL-13-K does not abrogate IL-4 and IL-13 production in this asthma model, but rather induces antibodies that neutralize these cytokines once released.

Neither single nor dual kinoid vaccination, however, altered eosinophil levels in circulation (Supplementary Fig. 9a), indicating that the reduced airway eosinophilia observed after dual vaccination was a consequence of reduced eosinophil recruitment to the lungs rather than systemic effects on eosinophil numbers or progenitors. In addition, while numbers of effector eosinophils were markedly reduced in the lungs of vaccinated mice (Fig. 1f, g), we observed no effect of the vaccine on the number of tissue-

resident regulatory eosinophils[23] (Supplementary Fig. 9b, c). The effect of the vaccine on leukocytes appeared to be mostly restricted to effector eosinophils and to a lesser extent B cells, as we observed similar levels of neutrophils, macrophages, and T cells in the BAL fluid of HDM-treated mice vaccinated with IL-4-K and IL-13-K or CRM197 as a control (Supplementary Fig. 10).

This HDM-induced asthma model also leads to pronounced peribronchiolar inflammation and mucus production[19]. Single or dual kinoid vaccination reduced peribronchiolar inflammation to a similar extent (Fig. 1i and scoring in Supplementary Fig. 11a), whereas IL-13 vaccination but not IL-4 vaccination was necessary to profoundly reduce mucus production (Fig. 1j and scoring in

**Fig. 1 Dual vaccination with IL-4-K and IL-13-K reduces features of chronic asthma. a** Protocol outline. Mice were vaccinated with IL-4-K and/or IL-13-K (or PBS or CRM[197] as controls), combined with the adjuvant SWE. At day 39, mice were sensitized and challenged with HDM or PBS, as indicated. **b, c** Anti-IL-4 (**b**) and anti-IL-13 (**c**) neutralizing capacity in sera collected at the indicated time points. Data show values from individual mice ($n = 12$/group) with bars indicating median, from a single experiment representative of three independent experiments. **d, e** Lung resistance (**d**) and elastance (**e**) in response to inhaled methacholine 24 h after the last HDM challenge. Data represent mean ± SEM from two independent experiments ($n = 8$ mice in IL-4-K + IL-13-K/HDM group; $n = 9$ mice in PBS/PBS and CRM[197]/HDM groups; or $n = 10$ mice in CRM[197]/PBS, IL-4-K/HDM, and IL-13-K/HDM groups). **f, g** Numbers of eosinophils in BAL fluid (**f**) and lung tissue (**g**) 24 h after the last HDM challenge. Data show values from individual mice ($n = 7$ mice for HDM/CRM[197] group and $n = 8$ mice for all other groups) with bars indicating mean ± SEM, from a single experiment representative of two independent experiments. **h** Levels of IL-5 in BAL fluid 24 h after the last challenge. Data show values from individual mice ($n = 8$/group pooled from two independent experiments) with bars indicating mean ± SEM. **i, j** Representative lung sections stained with hematoxylin and eosin (H&E; revealing leukocyte infiltration) (**i**), or periodic acid–Schiff (PAS; revealing mucus-producing goblet cells in dark purple) (**j**). Lower panels in **i** and **j** represent magnifications of the dashed areas. Lung sections are representative of each group ($n = 8$ mice/group). P values in **d** and **e** were calculated using two-way ANOVA followed by a Tukey posttest. **d**: ### < 0.0001, # = 0.0381, ***: 0.003 for IL-13-K/HDM, 0.0002 for IL-4-K + IL-13-K/HDM, <0.0001 for CRM[197]/PBS and PBS/PBS; **e**: ### < 0.0001, ** = 0.0019, ***: <0.0001 for all groups. P values in **f–h** were calculated using two-tailed Mann–Whitney U test (**f–h**) vs. CRM[197]/PBS (**f, g**), CRM[197]/HDM group (**h**), or indicated groups. Source data are provided in the Source data file.

Supplementary Fig. 11b), confirming the key role of IL-13 in mucus hypersecretion[6].

**Dual vaccination against IL-4 and IL-13 reduces IgE and mast cell numbers**. As IgE antibodies play an important role in allergic asthma[24,25], we next assessed the effects of IL-4-K and IL-13-K on IgE levels in the HDM-induced asthma model. Compared to PBS-treated (naive) mice, control immunization of CRM[197] in squalene-based adjuvant led to low, but detectable IgE levels in circulation but not, as expected, to detectable HDM-specific IgE (Fig. 2a, b). HDM-treated mice had higher total IgE and significant HDM-specific IgE levels in circulation. Single IL-4 and dual kinoid vaccination markedly reduced total and HDM-specific IgE levels, with more pronounced effects than anti-IL-13 vaccination (Fig. 2a, b), highlighting the prominent role of IL-4 in IgE production[6]. Noteworthy, HDM-treated mice also had elevated levels of HDM-specific IgG antibodies that were not affected by single or dual kinoid vaccination (Fig. 2c and Supplementary Fig. 12).

Mast cells are the main IgE effector cells in the lung[24]. In patients with allergic asthma, inhalation of an aeroallergen leads to crosslinking of membrane-bound allergen-specific IgE, inducing rapid release of mast cell mediators, such as histamine and tryptase[24]. As expected, the chronic i.n. exposure to HDM of the asthma model we use herein resulted in a marked increase in the numbers of lung mast cells, as compared to PBS-treated animals (Fig. 2d, e)[19]. This mast cell recruitment was abolished following IL-4-K or dual kinoid vaccination, and reduced ~2.5-fold following IL-13-K vaccination (Fig. 2d, e). Importantly, in addition to restoring basal numbers of mast cells in the lungs of HDM-treated mice, kinoid vaccination also markedly reduced IgE levels on mast cells (Fig. 2f, g), indicating that both vaccines induce successful mast cell "desensitization". This vaccination-induced desensitization was observed at the systemic level, as membrane-bound IgE levels were also markedly reduced on blood basophils (which also express the high-affinity IgE receptor FcεRI) and peritoneal mast cells upon vaccination with kinoids (Supplementary Fig. 13a, b).

Since dual IL-4-K/IL-13-K prophylactic vaccination prevented or strongly reduced all key features of HDM-induced asthma in mice, whereas single IL-4-K or IL-13-K vaccination affected only a subset of these features (Figs. 1 and 2), we assessed the efficacy of therapeutic vaccination only using dual IL-4-K/IL-13-K vaccination on mice with established asthma (Fig. 3a and Supplementary Fig. 14). Mice were preexposed to HDM for 3 weeks before the first injection of kinoids, and remained exposed to HDM once a week for a total of 15 weeks thereafter (Fig. 3a). Dual vaccination induced high levels of neutralizing

antibodies against both IL-4 and IL-13 whether mice were preexposed or not to HDM (Fig. 3b, c and Supplementary Fig. 15), suggestive of potential efficacy in a therapeutic setting. Indeed, dual therapeutic vaccination demonstrated a profound reduction in key features of asthma, including a ~2-fold reduction of total and HDM-specific IgE levels (Fig. 3d, e), of AHR to inhaled methacholine (Fig. 3f, g), of airway eosinophilia (Fig. 3h), and a ~6-fold reduction in mucus production (Fig. 3i, j).

**A human IL-4/IL-13 vaccine induces neutralizing responses in humanized mice**. Low interspecies similarity of IL-4 (~44%) and IL-13 (~55%) between mice and human would render mouse IL-4-K and IL-13-K highly immunogenic in humans, and less potent to generate neutralizing responses. We therefore developed and characterized kinoids eliciting an immune response against human IL-4 and IL-13 (hIL-4-K and hIL-13-K; Supplementary Fig. 16), and used mice humanized for IL-4, IL-13, and for their common receptor chain IL-4Rα (hIL-4/hIL-13[KI]; hIL-4Rα[KI] mice) by syntenic replacement of two mouse loci: that encoding Il4/Il13 and the second encoding Il4ra, with the corresponding human segment of DNA. These hIL-4/hIL-13[KI]; hIL-4Rα[KI] mice express the human genes in the place of the mouse gene, and thus cytokine receptor interactions in these animals model those of the human proteins (Supplementary Fig. 17). We confirmed that splenocytes from hIL-4/hIL-13[KI]; hIL-4Rα[KI] mice, but not from WT mice, release human IL-4 upon stimulation with PMA and ionomycin (Fig. 4a). We obtained similar results with human IL-13, and further showed that significant levels of human IL-13 can also be detected in BAL fluid from hIL-4/hIL-13[KI]; hIL-4Rα[KI] mice following chronic i.n. sensitization and challenge (Fig. 4b, c). We further confirmed expression of human IL-4Rα (and lack of expression of mouse IL-4Rα) by immunohistology in skin samples from hIL-4/hIL-13[KI]; hIL-4Rα[KI] mice (Fig. 4d, e). Finally, we showed that i.n. challenge with recombinant human IL-4 or human IL-13 leads to eosinophilia, lung inflammation, and mucus production in hIL-4/hIL-13[KI]; hIL-4Rα[KI] mice (Fig. 4f–j). Altogether, these data demonstrate that this new hIL-4/hIL-13[KI]; hIL-4Rα[KI] humanized mouse strain both produces and responds to hIL-4 and hIL-13.

Dual hIL-4-K/hIL-13-K vaccination induced neutralizing responses against both human IL-4 and IL-13 in all immunized mice (Fig. 5a–c and Supplementary Fig. 18). Confirming the efficacy of the human dual vaccine, mice showed a >2.5-fold reduction in circulating IgE readily detectable 5 weeks post primary vaccination (Fig. 5d), as well as a decrease in membrane-bound IgE on basophils (Fig. 5e). Mast cell number and IgE level were very low in the lungs of these mice that had not been previously exposed to allergens. However, we could efficiently

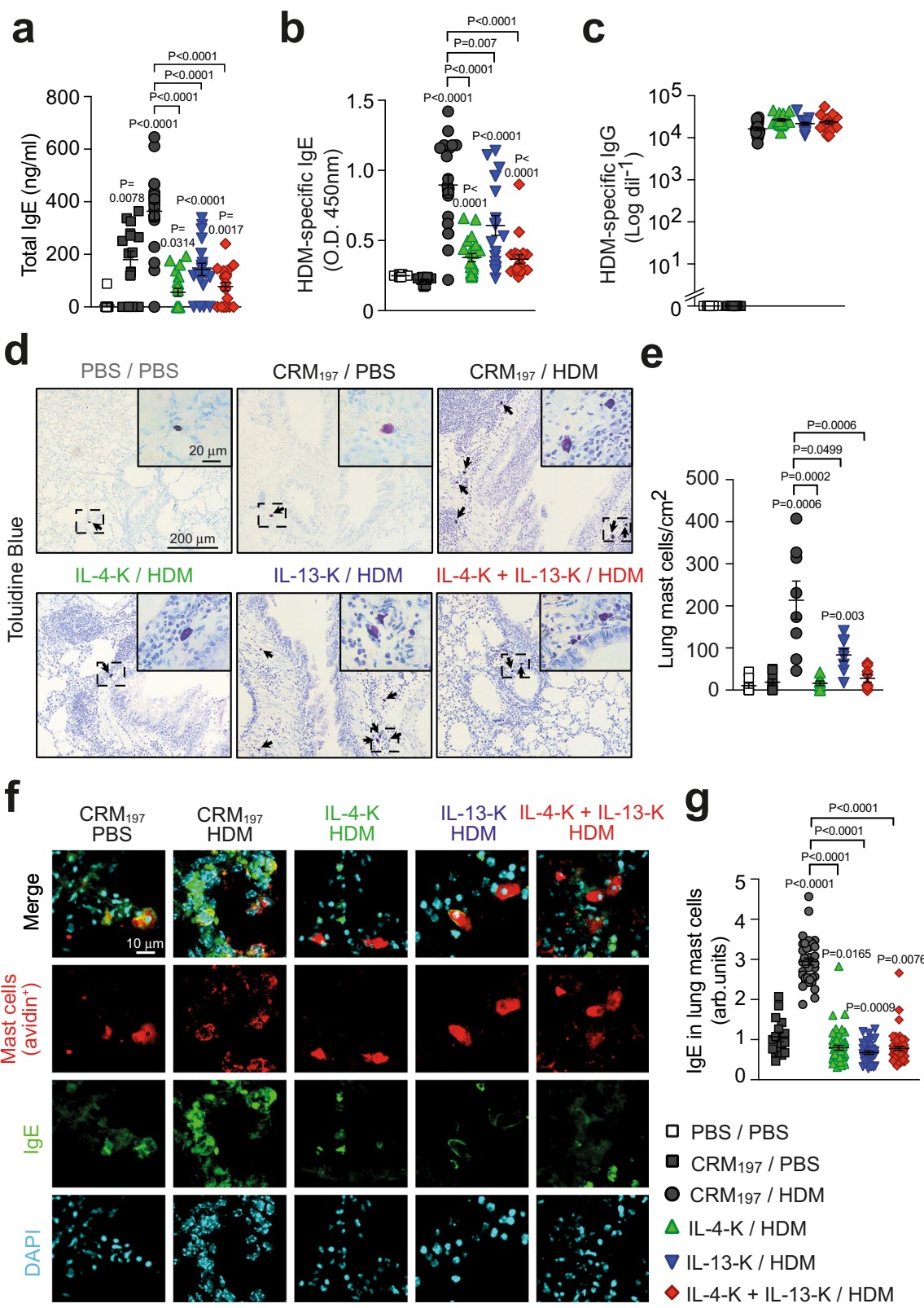

detect IgE in most skin mast cells from control hIL-4/hIL-13$^{KI}$; hIL-4Rα$^{KI}$ mice, and such IgE levels were reduced ~2.5-fold upon dual vaccination, with hIL-4-K and hIL-13-K (Fig. 5f, g).

## Discussion

Recent clinical data highlight the fact that IL-4 and IL-13 are important therapeutic targets in asthma[9]. However, targeting these cytokines or their receptors through the use of therapeutic mAbs is associated with high costs, and the need to perform frequent reinfusions in order to maintain clinical effects. Our current study provides a proof-of-concept that long-term neutralization of IL-4 and IL-13 can be achieved through vaccination with kinoids, which thus could represent a cost-effective alternative to therapeutic mAbs. We demonstrate that vaccination

**Fig. 2 Dual prophylactic vaccination with IL-4-K and IL-13-K prevents elevated IgE levels and lung mast cell numbers after HDM challenges. a–c** Levels of total IgE (**a**), HDM-specific IgE (**b**), and HDM-specific IgG (**c**) 24 h after the last HDM challenge. Results show values from individual mice with bars indicating mean ± SEM from $n = 12$ mice (PBS/PBS group), $n = 16$ mice (CRM$_{197}$/PBS group), $n = 19$ mice (IL-13-K/HDM group), or $n = 20$ mice (IL-4-K and IL-4-K + IL-13-K groups) pooled from two independent experiments. **d** Representative lung sections stained with toluidine blue, demonstrating mast cells (arrows) 24 h after the last HDM challenge. Insert represent magnifications of the dashed areas. **e** Quantification of toluidine blue-positive lung mast cells. Results show values from individual mice with bars indicating mean ± SEM from $n = 8$ mice. **f** Representative lung sections stained with avidin (which stains mast cells, red), anti-IgE (green), and DAPI (blue) 24 h after the last HDM challenge. **g** Quantification of IgE levels in avidin-positive lung mast cells. Results show values from individual avidin-positive mast cells analyzed in ear skin sections from $n = 4$ mice/group with bars indicating means ± SEMs. $P$ values were calculated using two-tailed Mann–Whitney $U$ test vs. PBS/PBS group (in **a**), CRM$_{197}$/PBS group (in **b**, **e**, and **g**) or vs. indicated groups. Source data are provided in the Source data file.

against IL-4 and IL-13 is well tolerated and protects against key features of chronic asthma in mice, including AHR, eosinophilia, and mucus overproduction, after both prophylactic or therapeutic vaccination protocols.

We observed different effects of prophylactic vaccination against IL-4 or IL-13 on asthma features, highlighting the fact that IL-4 and IL-13 can have important nonoverlapping functions in asthma[15]. In particular, vaccination against IL-4 had more pronounced effects on IgE levels and lung mast cell numbers than vaccination against IL-13. By contrast, AHR and mucus overproduction were reduced to a greater extent by the IL-13 vaccine. These results are in full agreement with previous observations in mice lacking IL-4 or IL-13[6,26–28]. Such nonoverlapping functions could be explained, at least in part, by the different receptor requirement for the two cytokines. The type 1 IL-4 receptor only recognizes IL-4, and is a heterodimer of IL-4Rα and the common gamma chain (γc)[6,29,30]. The type 2 IL-4 receptor binds both IL-4 and IL-13, and is a heterodimer of IL-4Rα and IL-13Rα1[6,29,30]. In addition, IL-13 (but not IL-4) also binds IL-13Rα2, which was long thought to function as a decoy receptor that limits the activity of IL-13[6,29,30]. However, evidence indicates that IL-13 signaling through IL-13Rα2 can induce production of TGF-β1[31].

In line with our data in mice, IL-13 also plays an important role in controlling airway reactivity in human asthma. Indeed, an increase in FEV1 (forced expiratory volume in 1 s) has been observed in clinical trials conducted with anti-IL-13 mAbs[14,32,33], especially when focusing on patients with biomarker evidence of type 2 asthma (e.g., high blood eosinophil counts or periostin concentrations, which is induced by IL-4 and IL-13, ref. [34]). However, LAVOLTA and STRATOS, two large clinical trials, failed to demonstrate an effect of lebrikizumab and tralokinumab (two anti-IL-13 mAbs) on asthma exacerbation[14,35]. Thus, we speculate that combined blockade of IL-4 and IL-13 pathways is probably also required in order to efficiently reduce most features of asthma in human, as we observed here in mouse models with the IL-4 and IL-13 vaccines. This provides a potential explanation for the superior clinical efficiency of dupilumab (which blocks both IL-4 and IL-13 signaling) over various therapeutic anti-IL-4 or IL-13 mAbs in asthma[13–15], and prompted us to focus on the dual vaccine for further evaluation in a therapeutic protocol.

The extent to which the protective effects of anti-IL-4 and IL-13 therapy in asthma reflects local blockade of the cytokines in the lung vs. systemic effects is still not fully understood. Interestingly, local delivery of an anti-IL-13 Fab fragment by nebulization has been tested in a model of allergic asthma in cynomolgus monkeys[36]. This Fab had moderate effects on BAL eosinophilia, but markedly reduced BAL IL-5 levels, which is in full agreement with our data obtained in HDM-exposed mice vaccinated with the IL-13-K. However, even though the anti-IL-13 Fab was delivered by nebulization, significant levels of the antibody fragment could be detected in circulation. By extension one may consider that systemic effects of the anti-IL-13-K

therapy are conceivable, even if we did not detect any significant systemic effects.

Levels of IL-4 and IL-13 neutralization obtained upon vaccination with kinoids will likely never reach levels observed directly after injection of high dose of a therapeutic mAb in human, or upon genetic ablation of IL-4 or IL-13 in mice. This was apparent in the therapeutic vaccination protocol in which IgE levels were reduced, but still detectable, in all vaccinated mice. Besides IgE, mice which fully lack IL-4 or IL-4Rα have markedly reduced IgG1 levels[37]. In addition, treatment of mice humanized for IL-4 and IL-4Rα with dupilumab also leads to important decrease in IgG1[38]. However, we found no difference in HDM-specific IgG1 levels between mice vaccinated with IL-4-K and IL-13-K or CRM$_{197}$ alone. This suggests that residual cytokine activity in mice vaccinated with kinoids might sustain IgG production, while reducing the pathogenic functions of IL-4 and IL-13 in asthma. STAT6 activation is a key step in IL-4/IL-13 signaling[6]. However, it is important to note that STAT6 can also be activated by other mediators, including TSLP[39] or cleaved fibrinogen[40]. This could also contribute to the residual allergic inflammation, IgE and IgG1 levels observed in mice vaccinated with IL-4-K/IL-13-K.

We also provided a proof-of-concept of the efficiency of kinoids targeting human IL-4 and IL-13 in mice humanized for these cytokines and their receptor IL-4Rα. These promising results will now need to be confirmed in clinical studies. In this regard, while IL-4 and IL-13 vaccines have never been tested in human, a kinoid targeting interferon alpha (IFN-α) has recently been tested in a phase 2b study in 185 adults with active systemic lupus erythematosus[41]. This IFN-α kinoid induced a neutralizing response against IFN-α in 91% of treated patients, with an acceptable safety profile[41]. In a follow-up study of the phases I/II trial, it was noted that most patients had a ≥10-fold reduction in anti-IFN-α antibody levels 1 year after the first injection of the vaccine[42]. Neutralizing antibodies were still detectable at last follow-up visit in 29% (6/21) patients who received the IFN-α kinoid (range of persistence: 0.45–4.2 years). Although the strength and duration of the antibody response induced by the human IL-4/13 kinoids will need to be determined in clinical trials, they may be similar to the responses observed with the IFN-α kinoids. In our preclinical models, we observed 60% persistence 1 year after primary immunization.

Besides their detrimental role in allergies, IL-4 and IL-13 also play important protective and immunoregulatory functions. In particular, these cytokines can induce host defense responses against helminths infections, and have been implicated in the promotion of anti-inflammatory and tissue repair phenotypes in macrophages[43–45]. Thus, even though we did not observe apparent side effects of the vaccines in a 1-year follow-up study in mice, further work is now required to evaluate whether residual IL-4 and IL-13 activity after vaccination with kinoids is sufficient to sustain protective type 2 immune responses. In particular, it will be important to assess whether separate or combined

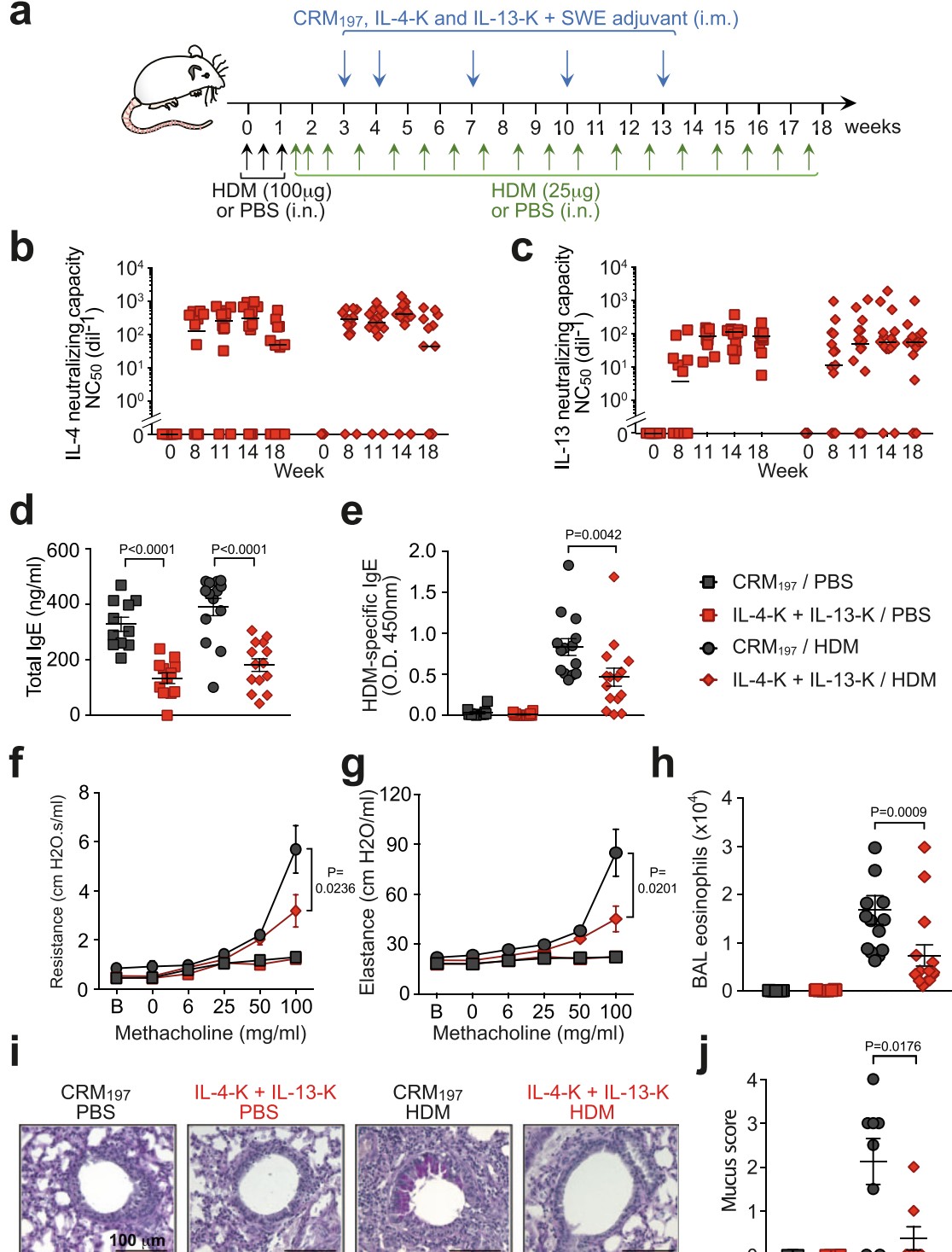

**Fig. 3 Dual therapeutic vaccination with IL-4-K and IL-13-K ameliorates chronic asthma. a** Protocol outline. Mice were sensitized and challenged with HDM extract (or PBS as a control), as indicated. After the third challenge, mice were vaccinated with IL-4-K and IL-13-K (or CRM197 as control), combined with the adjuvant SWE. **b, c** Anti-IL-4 (**b**) and anti-IL-13 (**c**) neutralizing capacity in sera collected at the indicated time points. **d, e**. Levels of total IgE (**d**) and HDM-specific IgE (**e**) 24 h after the last HDM challenge. **f, g** Lung resistance (**f**) and elastance (**g**) in response to inhaled methacholine 24 h after the last HDM challenge. **h** Eosinophil numbers in BAL fluid 24 h after the last HDM challenge. **i** Representative periodic acid–Schiff (PAS) staining of lung sections, demonstrating mucus-producing goblet cells (dark purple). **j** Quantification of mucus-producing goblet cells. Data show median (**b, c**) or mean ± SEM (**d–h, j**) from n = 4 (**j**), 7 (**f, g**), or 12 (**b–e**) mice in the PBS groups, and n = 8 (**j**) or 14 (**b–e**) mice in the HDM groups pooled from two independent experiments. Each symbol represents individual mice (**b–e, h, j**). P values were calculated using two-tailed Mann–Whitney U test in **b–e, h**, and **j**, or two-way ANOVA followed by a Tukey posttest in **f** and **g**. Source data are provided in the Source data file.

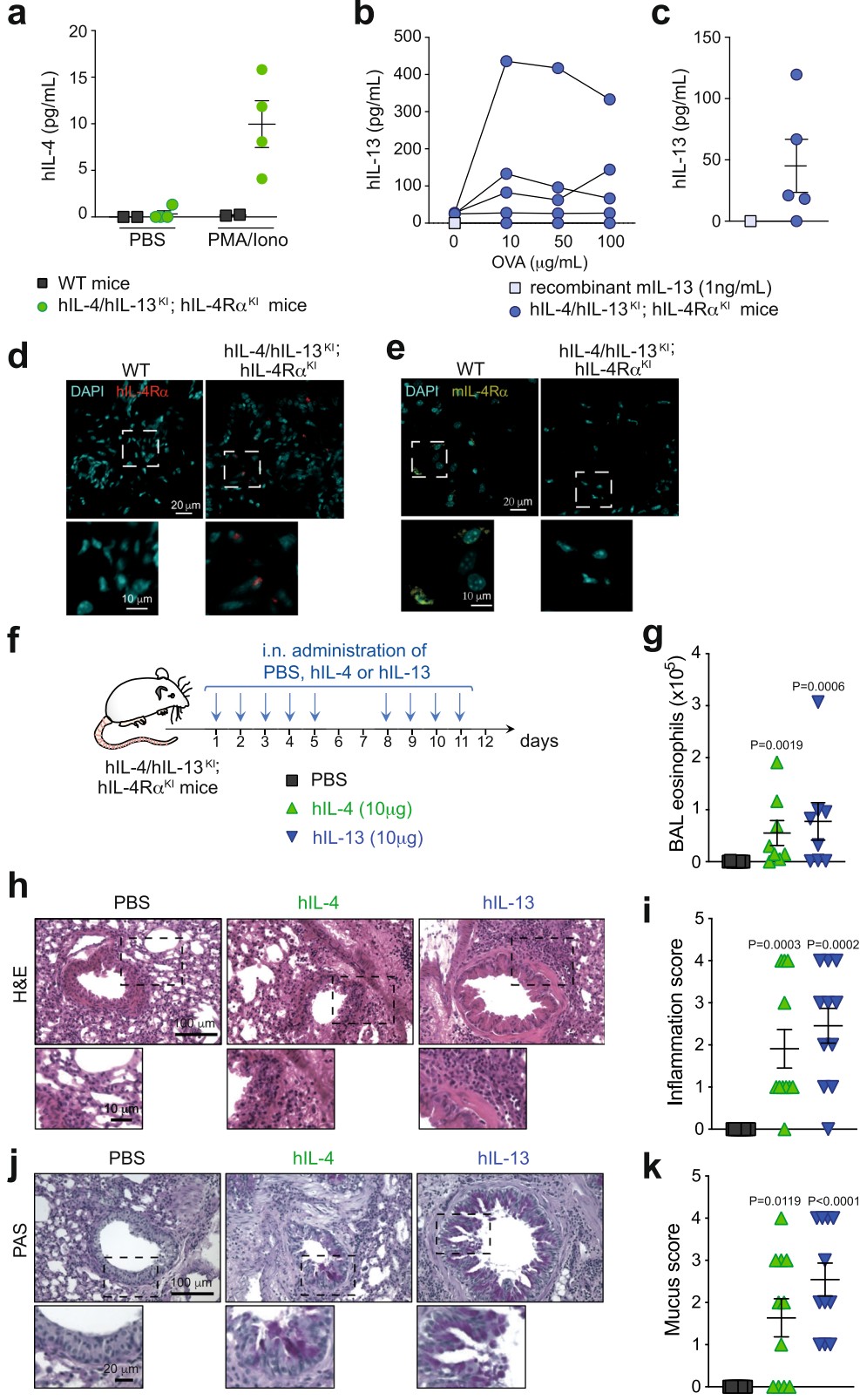

vaccination against IL-4 and IL-13 has any effect in models of infection with helminths. This question is particularly important since treatment with IL-4-K/IL-13-K in human may induce a long-lasting antibody response, for which the specific duration will need to be determined through clinical trials, that may not be desirable. Although data on the long-term effects of drugs targeting IL-4, IL-13, or other type 2 cytokines in human are still

limited, it is important to note that several large clinical studies are now available for the anti-IL-4Rα mAb with treatment periods from 52 to 96 weeks in both asthma and atopic dermatitis[9,46–50]. Overall, these studies demonstrate a good safety profile, which argues in favor of the feasibility of long-term targeting of IL-4 and IL-13 with a vaccine strategy. Interestingly, the most common side effect noted is conjunctivitis, although this

**Fig. 4 Characterization of hIL-4/hIL-13$^{KI}$; hIL-4Rα$^{KI}$ humanized mice. a** hIL-4 levels in the supernatant of splenocytes from $n = 2$ WT or $n = 4$ hIL-4/hIL-13$^{KI}$; hIL-4Rα$^{KI}$ humanized mice stimulated ex vivo with PMA (20 nM) **b** hIL-13 levels in the supernatant of splenocytes from hIL-4/hIL-13$^{KI}$; hIL-4Rα$^{KI}$ mice sensitized with ovalbumin (OVA), and stimulated ex vivo with OVA ($n = 5$ mice). **c** hIL-13 levels in BAL fluid from hIL-4/hIL-13$^{KI}$; hIL-4Rα$^{KI}$ mice sensitized and challenged with OVA ($n = 5$ mice). Samples were collected 24 h after the last OVA challenge. Data in **a**–**c** show values from individual mice with bars indicating mean ± SD (in **a** and **c**). **d, e** Representative staining of skin samples from WT or hIL-4/hIL-13$^{KI}$; hIL-4Rα$^{KI}$ mice with anti-mouse IL-4Rα (**d**) or the anti-human IL-4Rα mAb dupilumab (**e**) and DAPI. Staining skin sample are representative from three WT and six hIL-4/hIL-13$^{KI}$; hIL-4Rα$^{KI}$ mice from two separate experiments. **f** Protocol outline. hIL-4/hIL-13$^{KI}$; hIL-4Rα$^{KI}$ mice were challenged nine times intranasally with 10 µg recombinant hIL-4, hIL-13, or PBS as a control. **g** Eosinophil numbers in BAL fluid 24 h after the last challenge with hIL-4 or hIL-13. **h** Representative lung sections stained with hematoxylin and eosin (H&E; revealing leukocyte infiltration), 24 h after the last challenge with hIL-4 or hIL-13. **i** Scoring of leukocyte infiltration in H&E-stained lung tissue sections. **j** Representative periodic acid–Schiff (PAS) staining of lung sections, demonstrating mucus-producing goblet cells (dark purple). **k** Quantification of mucus-producing goblet cells. Data in **g**, **i**, and **k** show values from individual mice with bars indicating mean ± SEM from $n = 8$ mice per group, pooled from two independent experiments. $P$ values were calculated using two-tailed Mann–Whitney $U$ test. Source data are provided in the Source data file.

seems to be restricted to atopic dermatitis patients, as it is not observed for patients with moderate-to-severe asthma[9,46–48,51].

Altogether, our results indicate that long-term neutralization of both mouse and human IL-4 and IL-13 can be achieved through vaccination with kinoids. Dual vaccination could protect against key features of chronic asthma after both prophylactic or therapeutic vaccination protocols. These results pave the way for the clinical development of an efficient long-term vaccine against asthma and other IL-4- and IL-13-mediated allergic diseases, such as food allergies, atopic dermatitis, or chronic urticaria.

## Methods

**Mice.** Female BALB/cJRj mice at 5–6 weeks of age were purchased from Janvier Labs, and maintained in a specific pathogen-free facility at Institut Pasteur or Institut Jacques Monod. The *IL4RA/IL13/IL4* humanized mouse line (named hIL-4/hIL-13$^{KI}$; hIL-4Rα$^{KI}$ in the manuscript) was generated at University of North Carolina (USA) by syntenic replacement of two mouse loci: that encoding *Il4/Il13* and the second encoding Il4ra, with the corresponding human segment of DNA, as represented in Supplementary Fig. 17. The endogenous mouse *Il4ra* gene was deleted in ES cells with a replacement type targeting vector. Homologous recombination of this vector with the Il4ra locus resulted in deletion of a segment of the Il4ra locus extending from chr7:125,539,693–125,579,740 (GRCm38/mm10). The segment of the human IL4Rα locus extending from chr16:27,309,862–27,372,883 (GRCm38/mm10) human gene was inserted into the deleted locus by Cre-mediated recombination, using previously described methods[52]. Correct insertion of the human DNA segment into the deleted locus was verified by PCR analysis, Southern blot, and sequencing. A similar approach was used to replace the segment of the Il4/Il13 locus in ES cells extending from chr11:53,604,938–53,637,149 with the segment of the human IL-4/IL-13 locus extending from chr5:132,646,866–132,689,719. Mouse lines carrying the individual humanized loci were intercrossed to generate a line homozygous for the humanized loci. All animal care and experimentation were conducted in compliance with the guidelines and specific approval of the Animal Ethics committee CETEA (Institut Pasteur, Paris, France) registered under #170043, and by the French Ministry of Research. The protocol also received the authorization number EU0285 - Institut Jacques Monod PHEA - APAFiS - Autor. APAFiS #165.

**Synthesis and characterization of IL-4 and IL-13 kinoids.** Mouse IL-4 (214-14), human IL-4 (200-04), mouse IL-13 (210-13), and human IL-3 (200-13) were purchased from PeproTech. CRM$_{197}$ was purchased from Pfenex. IL-4 and IL-13 were modified with $N$-γ-maleimidobutyryl-oxysuccinimide ester (sGMBS; Thermo Fisher scientific, 22324), a maleimide-containing agent reacting with primary amines. Cytokines were dissolved in modification buffer (70 mM phosphate buffer, 150 mM NaCl, 5 mM EDTA, pH = 7.2) at 1 mg/ml. A solution of 10 mM of sGMBS was prepared and added to the cytokine at a 1:30, 1:20, or 1:10 molar ratio, for human IL-4, mouse IL-4, and human IL-13 or mouse IL-13 modification, respectively, and incubated during 1 h at room temperature (protected from light). Excess sGMBS was removed using a Zeba desalting spin column (Thermo Fisher scientific). Sulfhydryl moieties were introduced on the carrier protein CRM$_{197}$ with SATA ($N$-succinimidyl-$S$-acetylthioacetate; Sigma-Aldrich, A9043). CRM$_{197}$ was diluted in modification buffer at 2 mg/ml and a freshly prepared solution of 100 mM SATA (dissolved in DMSO) was added at a 1:80 molar ratio and incubated 30 min at room temperature (protected from light). Excess SATA was removed and buffer exchanged against modification buffer using a Zeba desalting spin column. SATA-modified CRM$_{197}$ was incubated with a solution of hydroxylamine hydrochloride (Thermo Fisher scientific, 26130) at a 50 mM final concentration, at room temperature for 2 h, protected from light. Excess hydroxylamine was removed and buffer exchanged against modification buffer using a Zeba desalting spin column.

After CRM$_{197}$ and IL-4 or IL-13 functionalization, protein content of each preparation was determined by Bradford assay according to manufacturer's instructions (Thermo Fisher scientific).

Functionalized CRM$_{197}$ was added to functionalized IL-4 or IL-13 at a molar ratio of 1:2 (for mouse IL-4) or 1:4 (for mouse IL-13, human IL-13, and human IL-4) and a final concentration of 0.4 mg/ml. The mixture was incubated 16 h at 4 °C, protected from light, and subsequently washed with fresh modification buffer using Zeba desalting spin column. Protein content was determined by Bradford assay. Resulting IL-4-K and IL-13-K were then 0.22 µm sterile filtered and stored at 4 °C. Kinoids were characterized using different in vitro methods. To analyze the profiles of the kinoids obtained, SDS–PAGE and western blots were performed with mouse IL-4 (using AF-404-NA as a detection antibody at 0.1 µg/ml, R&D systems), human IL-4 (AF-204-NA at 0.1 µg/ml, R&D systems), mouse IL-13 (AF-413-NA at 0.1 µg/ml, R&D systems), human IL-13 (AF-213-NA at 0.1 µg/ml, R&D systems), and with CRM$_{197}$ (AbD serotec, 3710-0956 at 860 ng/ml). Size-exclusion (SE)-HPLC were performed using a Bio SEC-5 column (2000 Å, 5 µm, 7.8 × 300 mm, Agilent) and a Bio SEC-3 column (300 Å, 3 µm, 7.8 × 300 mm, Agilent) in connected series. SE-HPLC analysis were performed in the isocratic mode at 1 ml/min with column temperature at 25 °C. After filtration (0.22 µm cut-off), samples were injected at 100 µl and analyzed at 280 nm. The total run time was 35 min.

To confirm coupling between the cytokines and the carrier protein, and to evaluate epitope preservation, antigenicity was analyzed by sandwich ELISA. Briefly, capture antibody (mouse monoclonal anti-diphtheria toxin, AbD serotec, 3710-0100) was coated overnight at 1 µg/ml. After each step, plates were washed three times with PBS 0.01% Tween 20. Then plates were blocked with casein 2% dissolved in PBS. Kinoid samples were added at 250 ng/ml and twofold serially diluted in 100 µl final volumes. After incubation, bound kinoids were detected using biotinylated anti-mouse IL-4 antibody (polyclonal goat IgG, R&D systems, BAF-404), biotinylated anti-human IL-4 antibody (polyclonal goat IgG, R&D systems, BAF204), biotinylated anti-mouse IL-13 antibody (polyclonal goat IgG, R&D systems, BAF-413), or biotinylated anti-human IL-13 antibody (polyclonal goat IgG, R&D systems, BAF213) at 250 ng/ml, and then revealed with streptavidin-HRP and an OPD substrate. The reaction was stopped adding 1 M H$_2$SO$_4$ after 30 min of OPD incubation at room temperature protected from light, and absorbance was subsequently recorded at 490 nm.

**Vaccination with kinoids.** Mice were immunized intramuscularly with IL-4-K and/or IL-13-K combined 1:1 (v:v) with SWE, a squalene-in-water emulsion adjuvant (Vaccine Formulation Laboratory, University of Lausanne, Switzerland) in PBS on days indicated in the protocol outlines on Figs. 1, 2, and 5, at two initial doses of 30 µg followed by boosts of 10 µg. As controls, groups of mice were injected with the same schedule with CRM$_{197}$ alone with two initial doses of 15 µg followed by boosts of 5 µg (these doses were defined based on the weight ratio of CRM$_{197}$ used to generate kinoids; as shown in Supplementary Figs. 1, 11, and 13), immunization with CRM$_{197}$ alone induced slightly higher levels of anti-CRM$_{197}$ antibodies than immunization with kinoids combined with SWE, or PBS alone. In the experiments depicted in Supplementary Fig. 5, mouse IL-4 and CRM$_{197}$ or mouse IL-13 and CRM$_{197}$ were also co-injected without prior conjugation, following the same immunization schedule.

**Mouse models of house dust mite-induced allergic asthma.** In the prophylactic protocol (Figs. 1 and 2), BALB/c mice were immunized with kinoids at days 0, 7, 28, and 55. Starting at day 39, allergic asthma was induced with i.n. exposure to crude HDM extracts (*D. farinae*; lot number: 307244, Greer laboratories). Lightly isoflurane-anesthetized mice were sensitized three times by i.n. exposure with HDM (100 µg in 30 µl PBS), or PBS alone as control, with a 3-day interval between each administration. Lightly isoflurane (3% in air)-anesthetized mice were challenged i.n. with 25 µg of HDM in 30 µl PBS, or PBS alone as a control, twice a week

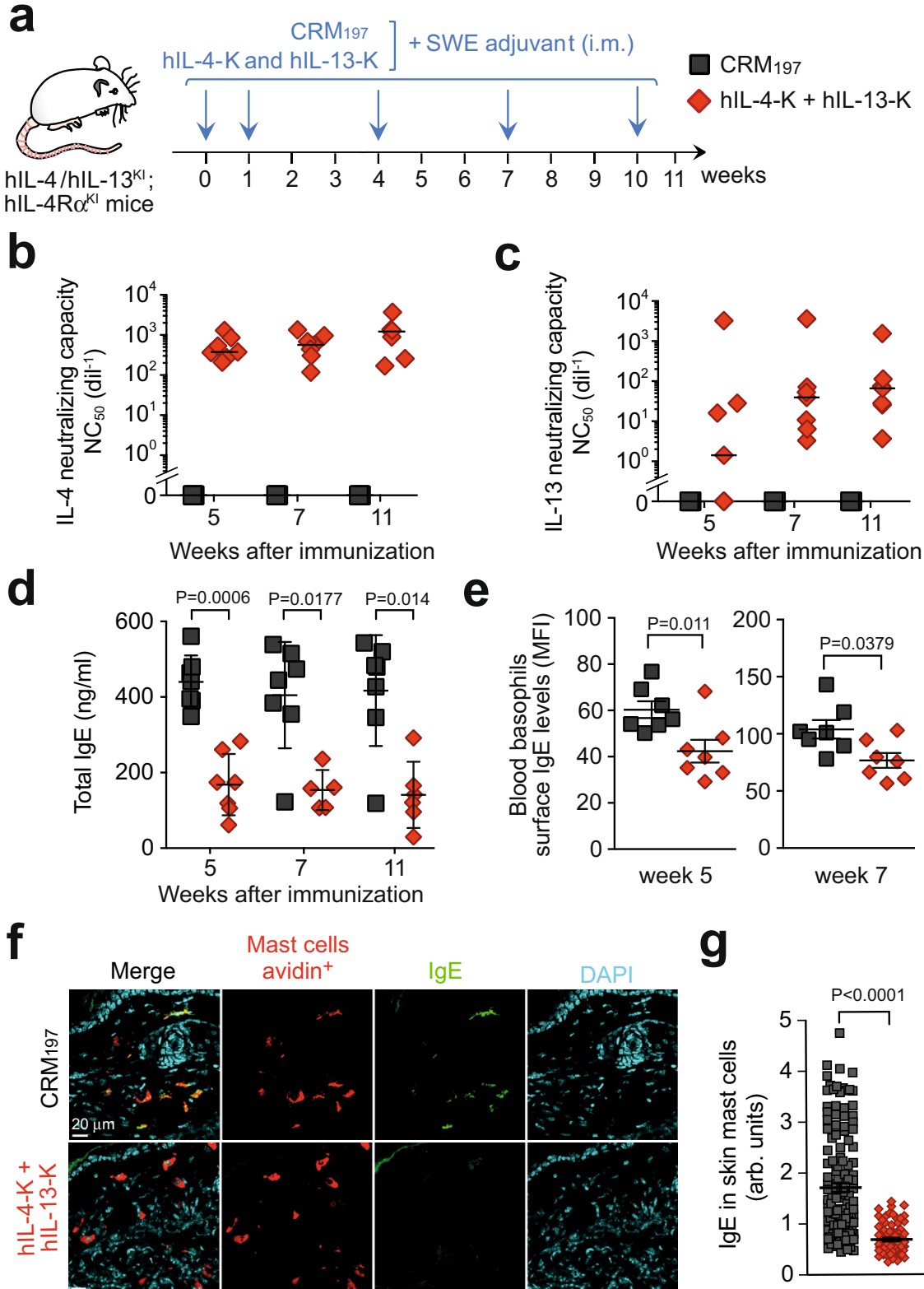

starting 3 days after the last sensitization, for a total of nine administrations. Mice were sacrificed 24 h after the last challenge with HDM or PBS.

In the therapeutic protocol (Fig. 3), BALB/c mice were sensitized three times with HDM (100 μg in 30 μl PBS), and PBS as control, with a 3-day interval between each administration. Lightly isoflurane (3% in air)-anesthetized mice were then challenged i.n. with 25 μg of HDM in 30 μl PBS, or PBS alone as a control, twice a week for a total of 18 challenges, starting 3 days after the last sensitization. Vaccination with mouse IL-4-K and mouse IL-13-K was initiated 4 days after the

third HDM challenge. Mice were sacrificed 24 h after the last challenge with HDM or PBS.

**Measurement of airway reactivity to methacholine**. Twenty-four hours after the last challenge with HDM or PBS, responses to aerosolized methacholine were measured using whole-body plethysmography (EMKA technologies), using IOX base 8c/RF-8a (0796) software. Responses to inhaled methacholine were assessed

**Fig. 5 Efficient vaccination with human IL-4 and IL-13 kinoids in IL-4/IL-13$^{KI}$; IL-4Rα$^{KI}$ humanized mice. a** Vaccination protocol outline. hIL-4/hIL-13$^{KI}$; hIL-4Rα$^{KI}$ mice were vaccinated with hIL-4-K and hIL-13-K in combination (or CRM$_{197}$ as control), combined with the adjuvant SWE. **b, c** Anti-human IL-4 (**b**) and anti-human IL-13 (**c**) neutralizing capacity in sera collected at the indicated time point. Results show values from individual mice ($n = 7$/group) with bars indicating medians. **d, e** Levels of total IgE (**d**) in sera or on the surface of blood basophils (**e**) at the indicated time points. Results show values from individual mice ($n = 7$/group) with bars indicating means ± SEMs. *, **, or ***: $P < 0.05$, 0.01 or 0.001 (two-tailed Mann–Whitney $U$ test). **f** Representative ear skin sections stained with avidin (which stains mast cells, red), anti-IgE (green), and DAPI (blue) 24 h after the last HDM challenge. **g** Quantification of IgE levels in avidin-positive mast cells. Results show values from individual avidin-positive mast cells analyzed in ear skin sections from $n = 7$ mice injected with CRM$_{197}$ and $n = 5$ mice injected with hIL-4-K and hIL-13-K, with bars indicating means ± SEMs. $P$ values were calculated using two-tailed Mann–Whitney $U$ test. Source data are provided in the Source data file.

by recording Penh over 5 min after each dose of aerosolized methacholine (baseline, 0, 3, 5, 7, and 14 mg/ml). Data were analyzed using Datanalyst software (DATA 4238). Invasive measurements were also performed in anesthetized, tracheostomized, mechanically ventilated mice using a FlexiVent software FlexiWare 8.0 (Scireq). Aerosolized methacholine was administered in increasing concentrations (baseline, 0, 6, 25, 50 and 100 mg/ml). Lung resistance ($R$) and tissue elastance ($E$) were computed by assuming a constant phase model.

**Intranasal challenges with IL-4 and IL-13.** hIL-4/hIL-13$^{KI}$; hIL-4Rα$^{KI}$ mice were exposed intranasally to 10 µg human IL-4 (Peprotech), 10 µg human IL-13 (Peprotech), or PBS as a control daily for 9 days. Twenty-four hours after the last administration, mice were sacrificed for analysis of BAL eosinophil numbers and lung inflammation.

**Flow cytometry analysis of leukocytes in blood, bronchoalveolar lavage fluid, peritoneal lavage, and lung tissue.** BALs were performed 24 h after the last challenge with HDM, hIL-4, or hIL-13 in anesthetized mice (187.5 mg/kg ketamine and 18.75 mg/kg xylazine). After semi-excision of the trachea, a plastic canula was inserted, and airspace was washed with 1 ml of PBS containing 2.6 mM EDTA and 2.5% (v/v) FBS. This operation was repeated for a total of three times.

For the analysis of leukocytes in lung tissue, right lung lobes were harvested 24 h after the last challenge with HDM, and transferred into gentleMACS C tubes (Miltenyi) containing lung dissociation kit (Miltenyi). Tubes were attached upside down on a gentleMACS dissociator (Miltenyi). After a washing step, red blood cells were lysed with ammonium chloride potassium (ACK) lysing buffer (Thermo Fisher scientific), and single-cell suspensions were 0.22 µm cut-off filtered. Single-cell suspensions of total right lung tissue and BAL fluid were stained with anti-CD45-FITC (clone # REA737, Miltenyi), anti-Ly6G-PE (clone # 1A8, BD Pharmingen), anti-CD11c-VB (clone # N418, Miltenyi), CD11b-VG (clone # REA592, Miltenyi), anti-Siglec-F-PECy7 (clone # REA798, Miltenyi), anti-B220-APC (clone # RA3-6B2, Miltenyi), and anti-CD3ε-APC (clone # 145-2C11, BD Pharmingen). Macrophages were gated as CD45$^+$, CD11c$^+$, Siglec-F$^+$, CD11b$^+$, B cells as CD45$^+$, CD11c$^-$, B220$^+$, T cells as CD45$^+$, CD11c$^-$, CD3ε$^+$, neutrophils as CD45$^+$, CD11c$^-$, B220$^-$, CD3ε$^-$, Ly6G$^+$, CD11b$^+$, and eosinophils as CD45$^+$, CD11c$^-$, B220$^-$, CD3ε$^-$, Ly6G$^-$, Siglec-F$^+$, SSC$^{high}$.

For intracellular staining of IL-4 and IL-13, single-cell suspensions from right lung lobes were first permeabilized with Perm/wash buffer (554723, BD Bioscience), according to the manufacturer's instructions. Cells were then stained with anti-CD4-FITC (clone # GK1.5, Miltenyi), anti-IL-4-PE (clone # BVD4-1D11 Miltenyi), and anti-IL-13-PE-Cyanine7 (clone # eBio13A, Fisher Scientific).

Blood was collected on heparin for analysis of eosinophils and basophils. Red blood cells were lysed with ACK lysis buffer (Thermo Fisher scientific). Cells were stained with anti-Siglec-F-PECy7 (clone # REA798, Miltenyi), anti-CD49b-APC (clone Dx5, eBioscience), and anti-IgE-FITC (clone # R35-72, BD Pharmingen). Blood eosinophils were gated as Siglec-F$^+$, SSC$^{high}$, and blood basophils as CD49b$^+$, IgE$^+$.

For analysis of peritoneal mast cells, 5 ml of PBS were injected into the peritoneal cavity and the abdomen was massaged gently for 20 s. Fluid containing peritoneal cells was collected and cells were stained with anti-c-KIT APC (clone # 2B8, Bioscience) and anti-IgE-FITC (clone # R35-72, BD Pharmingen). Peritoneal mast cells were gated as c-KIT$^+$, IgE$^+$. Samples were acquired on Miltenyi MACSQUANT 10 and 16. Data were analyzed using FlowJo 10.4.2 software. All FACS sequential gating strategies are presented in Supplementary Figs. 19 and 20.

We assessed levels of lung-resident regulatory eosinophils (rEos) in mice vaccinated with IL-4-K and IL-13-K or CRM$_{197}$. Right lung lobes were harvested and transferred into gentleMACS C tubes (Miltenyi) containing lung dissociation kit (Miltenyi). Tubes were attached upside down on a gentleMACS dissociator (Miltenyi). After a washing step, red blood cells were lysed with ACK lysing buffer (Thermo Fisher scientific). Single-cell suspensions were 0.22 µm cut-off filtered, and stained with anti-CD45-VB (clone # REA737, Miltenyi), CD125-PE (clone # T21, BD Pharmingen), Siglec-F-Alexa 647 (clone # E50-2440, BD Pharmingen), and propidium iodide (PI) solution (Miltenyi). rEos were gated as CD45$^+$, PI$^-$, CD125$^+$, Siglec-F$^{int}$, as previously described[23] (gating strategy is represented in Supplementary Fig. 9). Samples were acquired on BD LSRFortessa cell analyzer (BD Biosciences). Data were analyzed using FlowJo 10.4.2 software.

**Quantification of antibodies against mouse and human IL-4 and IL-13, and CRM$_{197}$.** The immunogenicity of the kinoids was assessed by evaluating antibodies against mouse IL-4, human IL-4, mouse IL-13, human IL-13, and CRM$_{197}$ in sera collected at different time points after vaccination. Mouse IL-4, human IL-4, mouse IL-13, human IL-13, or CRM$_{197}$ were coated and incubated overnight at 4 °C at 1 µg/ml. After each step, plates were washed three times with PBS Tween 20 0.01% (v/v). After blocking with casein 2% (w/v) in PBS, serum samples were added, a twofold serial dilution was conducted starting at 500 dil$^{-1}$ (diluted in PBS, casein 1% (w/v) and Tween 20 0.01% (v/v)). After 90 min of incubation at 37 °C, bound antibodies were detected with HRP-conjugated anti-mouse IgG (Invitrogen), and plates were revealed using an OPD substrate. Reaction was stopped with 1 M H$_2$SO$_4$ after 30 min of OPD incubation at room temperature protected from light, and absorbance was subsequently recorded at 490 nm. Samples were analyzed starting at dilution 500 dil$^{-1}$ up to 256,000 dil$^{-1}$, except for pre-immune sera analyzed only at 500 dil$^{-1}$. The titers were defined using Microsoft Excel 16.16.23 as the dilution of the serum, where 50% of the ODmax minus OD of corresponding pre-immune sample in the assay was reached. Titers were expressed as serum dilution factors (dil$^{-1}$). The limit of titer quantification is the lowest dilution tested in the assay: 500 dil$^{-1}$.

**Assessment of the neutralizing capacity against IL-4 and IL-13 in sera from vaccinated mice.** Neutralizing capacities of the anti-mouse IL-4 antibodies were evaluated using CTLL-2 cells proliferation assay (ECACC, ref. 93042610, batch number: 12K006.). Cells were grown in presence of human IL-2 (Sigma-Aldrich; 10 ng/ml). For neutralization bioassays, human IL-2 was replaced by mIL-4 (Peprotech; 2 ng/ml). Dilution series of serum samples from mice vaccinated with IL-4 kinoids were mixed with mouse IL-4 (2 ng/ml). After 1 h incubation, 20,000 CTLL-2 cells were added to preincubated samples. After 48 h, cell viability was quantified by MTS/PMS assay (Promega), according to the manufacturer's instructions. Neutralizing capacities of anti-human IL-4, anti-mouse IL-13, and anti-human IL-13 antibodies were evaluated using a HEK-Blue IL-4/IL-13 reporter gene cell line bioassay (InvivoGen, hkb-il413, batch number: X14-37-01), adapted from the manufacturer's instructions. When activated with human IL-4, mouse IL-13, or human IL-13, this cell line produces secreted embryonic alkaline phosphatase, which can be quantified using QUANTI-Blue medium (InvivoGen). Briefly, dilution series of serum samples from mice vaccinated with kinoids were mixed with mouse IL-13, human IL-13 (PeproTech; 2 ng/ml), or human IL-4 (PeproTech; 0.25 ng/ml), and then added to 40,000 HEK-Blue IL-4/IL-13 cells. After 24 h, supernatants were harvested and mixed with QUANTI-Blue. The IL-4/IL-13 neutralizing capacity 50 (NC$_{50}$) result was expressed as the serum dilution factor (dil$^{-1}$) neutralizing 50% of IL-4 or IL-13 activity, using Microsoft Excel 16.16.23.

**Quantification of total IgE levels and HDM-specific IgE and IgG.** Total IgE levels were quantified using a commercial ELISA kit (E90-115; Bethyl Laboratories), according to the manufacturer's instructions. HDM-specific IgE in sera were measured by ELISA, using a protocol adapted from refs. [53,54]. First, HDM was biotinylated using NHS-PEG4-biotin (molar ratio NHS-PEG4-biotin/HDM: 20/1) in phosphate buffer 70 mM, NaCl 150 mM, pH 7.2. The mixture was allowed to react for 30 min, at room temperature with a modification concentration of 0.6 mg/ml. Excess NHS-PEG4-biotin was removed using Zeba desalting spin column. Protein content was determined by Bradford assay. Effective HDM biotinylation was confirmed by direct ELISA: plate was coated with HDM-biotin, detected with poly-HRP streptavidin and revealed using an OPD substrate. HDM-specific IgE were detected by ELISA. Goat polyclonal anti-mouse IgE antibody (STAR110, Bio-rad) was coated and incubated overnight at 4 °C at 2 µg/ml in PBS. After each step, plates were washed three times with PBS Tween 20 0.01% (v/v). After blocking with casein 2% (w/v) in PBS for 90 min at 37 °C, serum samples were added at a 1:50 dilution (diluted in PBS, casein 1% (w/v), Tween 20 0.01% (v/v)) and incubated for 2 h at 37 °C. Then, HDM-biotin (prepared as described hereabove) was added at a 1:400 final dilution and incubated for 2 h at 37 °C. Bound HDM-specific IgE antibodies were detected with poly-HRP streptavidin (N200; Thermo Fisher scientific; dilution 1:10,000, 60 min at 37 °C incubation) and plates were revealed using an OPD substrate. Reaction was stopped with 1 M H$_2$SO$_4$ after 30 min of OPD

incubation protected from light, at room temperature and absorbance was subsequently recorded at 490 nm.

HDM-specific IgG, IgG1, and IgG2a levels in sera were measured by ELISA. HDM was coated in 96-well plates and incubated overnight at 4 °C at 5 µg/ml. After each step, plates were washed three times with PBS Tween 20 0.01%. After blocking with BSA 1% in PBS for 90 min, serum samples were added, a twofold serial dilution was conducted starting at 1:2000 (in PBS, BSA 0.5%, Tween 20 0.01%), 1:4000 (in PBS, BSA 1%), or 1:500 (in PBS, BSA 1%) for IgG, IgG1, or IgG2a respectively. After 90 min of incubation, bound antibodies were detected with HRP-conjugated anti-mouse IgG (Invitrogen) at 1:5000, goat HRP-conjugated anti-mouse IgG1 (Southern Biotech) at 1/8000, or goat HRP-conjugated anti-mouse IgG2a (Southern Biotech) at 1:8000, and plates were revealed using an OPD substrate. Reaction was stopped with 1 M $H_2SO_4$ after 30 min of incubation at room temperature, and absorbance was subsequently recorded at 490 nm.

**Quantification of mouse IL-5, human IL-4, and human IL-13 levels.** Mouse IL-5 levels were quantified in BAL fluid collected 24 h after the last challenge with HDM, using MACSPlex Cytokine (Miltenyi), according to the manufacturer's instructions. For quantification of human IL-4 (hIL-4), splenocytes from naive hIL-4/hIL-13$^{KI}$; hIL-4Rα$^{KI}$ mice were harvested, and cells were stimulated overnight with PMA/ionomycin (20 nM: 1 µM) or PBS as control at 37 °C. For quantification of hIL-13, hIL-4/hIL-13$^{KI}$; hIL-4Rα$^{KI}$ mice were sensitized with 50 µg ovalbumin (OVA) and 2 ml aluminum hydroxide gel (vac-alu-250, Invivogen) at days 0 and 7, and challenged intranasally with 10 µg OVA at days 21–24. BAL fluid was collected 24 h after the last challenge with OVA, as described above. Splenocytes were harvested and stimulated with OVA (10, 50, and 100 µg/ml) for 5 days at 37 °C. hIL-4 and hIL-13 were quantified by ELISA in supernatants from splenocytes or in BAL fluid samples. Mouse anti-hIL-4 (MAB304; clone #3007, R&D systems) or hIL-13 antibody (MAB213; clone #32116, R&D systems) were coated and incubated overnight at 4 °C at 1 µg/ml in PBS. After each step, plates were washed three times with PBS Tween 20 0.01% (v/v). After blocking with BSA 1% (w/v) in PBS for 60 min at 37 °C, serum samples and hIL-4 or hIL-13 as controls starting from 1 ng/ml were added (diluted in RPMI $+$ 10% FBS or PBS) and incubated for 90 min at room temperature. Then, hIL-4 or hIL-13 were detected with either biotinylated goat polyclonal anti-hIL-4 (BAF204, R&D systems) or hIL-13 (BAF213, R&D systems) antibodies at 1 µg/ml for 90 min at room temperature, followed by Strep-HRP (R&D systems) detection for 60 min at room temperature. Plates were revealed using an OPD substrate. Reaction was stopped with 1 M $H_2SO_4$ after 30 min of OPD incubation protected from light at room temperature. Absorbance was subsequently recorded at 490 nm.

**Lung histology.** Left lungs were excised from mice postmortem, fixed with 4% paraformaldehyde (PFA) for 24 h at room temperature, and preserved in 70% ethanol. Longitudinal sections were done and stained with hematoxylin and eosin (H&E; for assessment of leukocyte infiltration), periodic acid–Schiff (PAS) staining (for assessment of goblet cells hyperplasia and mucus production) or toluidine blue (for quantification of mast cell numbers; all from Sigma). The severity of inflammation on H&E-stained lung sections was graded semi-quantitatively in a blind manner for the following features: 0: normal, 1: few cells, 2: a ring of inflammatory cells, 1 cell layer deep, 3: a ring of inflammatory cells 2–4 cells deep, and 4: a ring of inflammatory cells of >4 cells deep (adapted from ref. [55]). The extent of mucus production was also quantified in a blind manner on PAS-stained lung sections by a score according to the percentage of goblet cells in the epithelial cells[56]: 0: no goblet cells, 1: <25%, 2: 25–50%, 3: 50–80%, and 4: >80%. Mast cell quantification on toluidine blue-stained lung sections was performed using Zen Software 3.1 blue edition.

**IHC section and acquisition.** For lung and ear skin IHC, the bigger lobe from the right lung or the ears were excised postmortem fixed with 1% PFA for 24 h at room temperature. Tissue dehydration was performed in sucrose gradient baths (10, 20 and 30%), then tissues were embedded in OCT compound.

A total of 8-µm thick sections of lung tissue sections were treated using a heat-induced epitope retrieval method, as previously described[57]. Tissue sections were blocked and permeabilized with PBS 0.5% (w/v) BSA (Sigma-Aldrich), 0.3% Triton X-100 (Merck) for 1 h at room temperature. For staining of lung IgE$^+$ mast cells, permeabilized tissue sections incubated with an anti-IgE antibody coupled to AF488 (Biolegend, UK), at 10 µg/ml overnight at 4 °C in the dark. Tissue sections were then washed three times in PBS 0.5% (w/v) BSA, and incubated with DAPI (Thermo Fisher scientific) and avidin-sulforhodamine at 5 µg/ml (Sigma-Aldrich; to stain mast cells[58]) in PBS 0.5% (w/v) BSA, for 2 h in the dark. Finally, samples were mounted in Mowiol medium (Sigma-Aldrich) and sealed with nail polish. A total of 512 × 512 pixels Z-stack images were acquired using a confocal microscope SP8 (Leica Microsystems) equipped with a HC PL APO CS2 with 20× NA 0.75 dry or a HC PL APO CS2 60×/NA 1.40 oil objective. In our study, a digital zoom of three or six was applied for 60× or 20× objective, respectively. A maximum intensity projection (MIP) was used for generating 2D images. Individual mean intensity analysis of anti-IgE staining on avidin-sulforhodamine-positive mast cells was performed using ImageJ (NIH). For staining of lung IL-4$^+$ type 2 ILC2, permeabilized skin sections were incubated overnight at 4 °C in the dark AF488-

coupled anti-KLRG1 (clone 2F1, BD Pharmingen), APC-coupled anti-IL-4 (clone 11B11, Thermofisher Scientific), and AF532-coupled anti-CD3 (clone 17A2, eBioscience). Sections were then extensively washed, incubated with DAPI for 1 h at room temperature in the dark and mounted in Mowiol mounting medium. Images of 512 × 512 pixels were acquired using Leica Microsystems SP8 confocal laser-scanning microscope equipped with a HC PL APO CS2 with 20× NA 0.75 dry. Images were processed using Zen Software 3.1 blue edition (Zeiss) and ImageJ (1.53c) software. CD3$^-$KLRG1$^+$IL-4$^+$ ILC2 were counted on images of four to six consecutive microscopic fields from each mouse.

A 8-µm thick sections of ear skin sections were blocked and permeabilized with PBS supplemented with 0.5% (w/v) BSA (Sigma-Aldrich) and 0.1% saponin (Sigma-Aldrich), for 30 min at room temperature, and incubated with an anti-IgE antibody coupled to Alexa Fluor (AF) 488 (Biolegend, UK), at 10 µg/ml overnight at 4 °C in the dark. Tissue sections were then washed three times in PBS 0.5% (w/v) BSA and 0.1% saponin, and incubated with DAPI and avidin-sulforhodamine in PBS 0.5% (w/v) BSA and 0.1% saponin, for 2 h at room temperature in the dark. Finally, samples were mounted in Mowiol medium (Sigma-Aldrich). A total of 512 × 512 pixels Z-stack images were acquired using a confocal microscope LSM710 (Zeiss) equipped with a HC PL APO 40×/NA 1.30 oil objective. A MIP was used for generating 2D images. Individual mean intensity analysis of anti-IgE staining on avidin-sulforhodamine + mast cells was performed using ImageJ (NIH).

For staining of mouse and human IL-4Rα in wild-type and hIL-4/hIL-13$^{KI}$; hIL-4Rα$^{KI}$ mice, skin tissue sections were blocked with PBS supplemented with 1% (w/v) BSA (Sigma-Aldrich), for 1 h at room temperature. Skin sections were then incubated with the anti-human IL-4Rα mAb dupilumab labeled in-house with AF647 (using Alexa Fluor 647 Protein Labeling Kit #A20173, Thermo Fisher, according to manufacturer's instructions) at 20 µg/ml, or a rat-anti-mouse IL-4Rα (clone mIL4R-M1, BD Pharmingen) at 10 µg/ml overnight at 4 °C in the dark. Tissue sections were then washed three times in PBS 1% (w/v) BSA, and incubated with goat anti-rat AF594 (1:150 in PBS 1% (w/v) BSA) for 1.5 h, followed by washing steps again three times in PBS 1% (w/v) BSA. Finally, tissue sections were incubated in 1:5000 with DAPI in PBS 1% (w/v) BSA for 15 min at room temperature in the dark, rinsed in PBS 1% (w/v) BSA, and mounted using DABCO mounting medium and nail polish. To image mouse IL-4Rα marked with AF594; 1524 × 1524 pixel Z-stacks were acquired using a LSM710 (Zeiss) confocal microscope equipped with 20×/0.8 plan apo. To image human IL-4Rα marked with AF647; 1024 × 1024 pixel Z-stacks were acquired using a SP8 (Leica) confocal microscope equipped with 20×/0.8 plan apo. A MIP was used for generating 2D images in ImageJ 1.53c.

**Statistical analysis.** Statistical significance was determined using the unpaired Student's $t$ test (unpaired Mann–Whitney $U$ test) or test two-way ANOVA followed by a Tukey posttest. $P \le 0.05$ was considered statistically significant. Calculations were performed using the Prism® 7.0 software program (GraphPad 7.0 Software).

**Reporting summary.** Further information on research design is available in the Nature Research Reporting Summary linked to this article.

## Data availability
All data generated or analyzed during this study are included in this published article (and its Supplementary information files). Source data are provided with this paper.

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

## Acknowledgements

The authors acknowledge Dr. Livia Brunner and Dr. Nicolas Collin from the Vaccine Formulation Laboratory (University of Lausanne, Switzerland) for the provision of SWE adjuvant, and thank Dr. Thomas Marichal (GIGA Institute, Liege, Belgium) for helpful advices on the gating strategy to identify lung-resident regulatory eosinophils. We thank Lhorane Lobjois for technical assistance at the Cell Imaging Facility of Infinity (Inserm UMR1291, Toulouse, France) and Dr. Muriel Pichavant (Institut Pasteur, Lille, France) for technical advices on airway measurements. Requests regarding hIL-4/IL-13; hIL-4Rα^KI humanized mice should be made to Dr. Beverly Koller (University of North Carolina, USA). E.C. is the recipient of a CIFRE Ph.D. fellowship. B.B. was supported partly by a stipend from the Pasteur—Paris University (PPU) International Ph.D. program, and a fellowship from the French "Fondation pour la Recherche Médicale FRM". N.G. acknowledges funding from the European Research Council (ERC-2018-STG, No. 802041) and the INSERM ATIP-Avenir program, B.H.K. acknowledges funding from the National Institutes of Health (NIH) HL093735 and an Award from the American Asthma Foundation. L.L.R. acknowledges support from the ATIP-Avenir program, and P.B. from the European Research Council (ERC)–Seventh Framework Program (ERC-2013-CoG 616050). This work was supported by a grant from the French National Research Agency ANR-18-CE18-0023 "AllergyVACS", NEOVACS, the Institut National de la Santé et de la Recherche Médicale (INSERM) and the Institut Pasteur.

## Author contributions

Experimental design, E.C., R.B, G.G.-V., and L.L.R; investigation, E.C., R.B., B.B., J.B., J.S., R.H., J.B.J.K., N.C., F.C., S.H., D.H., L.G., A.M., A.L., F.H., N.G., and L.L.R; generation of

hIL-4/hIL-13$^{KI}$; hIL-4Rα$^{KI}$ mice, J.N.S. and B.K; formal analysis, E.C., R.B., P.B., N.C., F.C., S.H., V.S., R.H., J.K., N.G., G.G.-V., and L.L.R.; writing (original draft), E.C. and L.L.R.; and writing (review and editing), all authors.

## Competing interests

E.C., R.B., P.B., N.C., V.S., G.G.-V., and L.L.R. are inventors on patent related to this work (WO2019228674 (A1)), and/or related to Kinoid technology. E.C., R.B., J.B., N.C., F.C., S.H., V.S., and G.G.-V. are currently or were previously employees of NEOVACS and company stocks owners. The other authors declare no competing interests.
