## [Peer Review File · Nature Communications]

REVIEWER COMMENTS

Reviewer #1 (Airway inflammation, type 2 immunity) (Remarks to the Author):

The authors demonstrate convincingly that vaccination against IL-4 and IL-13 can be achieved in wild type mice that are tolerized to these cytokines and furthermore that such vaccination is highly effective at preventing and attenuating HDM-mediated allergic airway disease. The results furthermore indicate that neutralizing antibody levels are quite durable over periods of up to a year. The experimental design, data description and analysis, and conclusions are outstanding. The figures are also of outstanding quality.

Major remaining questions and concerns:

1. How well can humans in general be vaccinated against these cytokines?
2. Will the vaccination approach be effective in persons at the extremes of age-young children and the elderly?
3. What is the long-term effect of having no IL-4/13 and no means of readily correcting this deficiency if needed?

Only human studies can of course answer these questions, but the discussion is quite brief and could be expanded to consider these issues not already addressed.

Other questions/issues possibly useful for considering in either the introduction or discussion:

1. Why did the authors not choose a simpler vaccination approach by choosing IL-4Ra as the antigen?
2. While the approach may be very helpful in asthma, it is likely to be equally or even vastly more useful in other conditions such as chronic urticaria; severe allergic rhinitis; chronic idiopathic anaphylaxis; food allergies; and chronic rhinosinusitis.
3. The approach ultimately works because it inhibits activation (by IL-4 and IL-13) of the transcription factor STAT6. Limiting the effectiveness of the approach-same concern with dupilumab-is that there are other soluble factors distinct from IL-4/13 that activate STAT6 in the allergic context, including TSLP (Han et al., J Invest Derm 2014) and cleaved fibrinogen (FCPs; Millien et al., Science 2013)

Other comments:

1. Line 191: humans, not men.
2. Line 56: though, not tough

David B. Corry, M.D.

Reviewer #2 (Allergy, asthma) (Remarks to the Author):

The work by Conde and colleagues reports on the effects of autovaccination for IL-4 and/or IL-13 in a murine model of allergic asthma to house dust mite. Using this elegant methodology, they were able to demonstrate the efficacy in this preclinical model that is relevant to targeted therapy of type 2 asthma, in particular with allergic phenotype, with the approved biologic dupilumab (anti-IL-4Ra mAb). Only partly overlapping were observed when comparing IL-4 and IL-13 (and dual) blockade, consistently with previous studies. The study was very well designed, using a relevant experimental model and readouts. Some issues could however be addressed to further elucidate the mechanisms of action.

Specific comments.

1. It is quite clear from the data that downstream targets of IL-4 and/or IL-13 are effectively altered following the vaccination protocol. It should be studied whether it also affected the functionality of type 2 T cells, either Th2 cells or ILC2 as well as the production of IL-5 which is shared by those cells and mediates tissue influx and activation of eosinophils. In addition, it would be interesting to look whether it targets mainly effector eosinophils and not the recently reported regulatory eosinophils in the lung.

2. The authors documented by invasive lung function tests that the protocol affected AHR, at least in the prophylactic setting. therefore, in addition to mucus production, it should be confirmed whether the treatment approach (in particular targeting IL-13 or both IL-4/IL-13) did alter smooth muscle hypertrphy that is classically seen (at least in human asthma, but also in chronic experimental asthma).

3. It would be extremely interesting to see whether this strategy is applicable in human asthma patients, although the reviewer aknowledge this will be the next step through an early phase trial. It could be however important, in order to get some translational applicability, to discuss the contrast between the efficacy of IL-13 vaccination in mice and the failure of anti-IL-13 mAbs in human (severe) asthma. Similarly, it should be discussed to what extent findings could be extrapolated to type 2 non allergic asthma (which may be also very eosinophilic, as some cases associated with nasal polyposis), notably because dupilumab efficacy does not rely on the allergic nature of asthma. Could the Authors comment on those translational aspects?

Charles Pilette, MD PhD.

Reviewer #3 (Asthma, allergy) (Remarks to the Author):

The manuscript by Conde et al. describes the effects of a dual vaccination against IL-4 and IL-13 in a mouse model of allergic asthma. As worded by the authors, the rationale for this work is that "dupilumab - a monoclonal antibody (mAb) against IL-4Ra that blocks both IL-4 and IL-13 signaling - is efficient at decreasing the rate of severe exacerbations, and at improving lung function in patients with moderate-to-severe asthma. However, use of this (or any other) mAb in chronic asthma is limited by high cost and the need to perform injections overs years to lifelong." This is a somewhat biased view. Costs are currently high but there is no reason they could not be lowered in the future. Most importantly, though, the need for lifelong therapy has the positive flipside that it makes it possible to stop treatment as needed – an option that does not appear to be available in the authors' approach. Type 2 immunity (both innate and adaptive) has protective, tissue repair-inducing properties, not to mention its protective effects against parasites and possibly viruses such as SARS-CoV-2. In this context, long-lasting elimination or even just suppression of type 2 responses by vaccination is a questionable, concerning goal. A more nuanced and practical alternative to addressing the asthma problem may be to prevent its inception by engaging natural regulatory/balancing mechanisms.

Some specific issues:

- The statement that "type 2 inflammation characterized by high levels of cytokines such as interleukin-4 (IL-4) and IL-13, high levels of IgE antibodies, and airway eosinophilia occurs in approximately 50 % of patients with asthma" is overall true but incomplete. Except for high IgE, these responses are essentially tissue-specific, not systemic. Therefore, it is not clear that a systemic suppressive approach would be optimal.
- Given the topic of this work, the authors may want to discuss the ligand/receptor interaction patterns that underpin the differential asthma-promoting effects of IL-4 and IL-13 despite their shared receptor.

- As mentioned above, reading that "Such neutralizing capacity could still be detected in more than 60 % of the mice over one year after primary immunization" is perplexing, given the life span of a mouse – especially because there is no switch off mechanism for this approach.
- For the reasons repeatedly mentioned above, the proposed vaccination strategy might be translationally significant (perhaps!) only as a treatment, not as prevention. However, the effects of dual vaccination in the therapeutic model are tenuous, especially for AHR.
- The authors should directly investigate, not just discuss, how their vaccination strategy affects protective type 2 responses, both innate and adaptive.
- The dissection of the differential effects of IL-4 and IL-13 in asthma pathogenesis, while interesting, is not novel.
- The humanized mice used in these experiments need to be described in some detail. In particular, it is important to specify whether they are knock-ins (as their denomination acronym might indicate) or they still carry the endogenous mouse genes. Result interpretation will be deeply affected in the latter case.

REVIEWER COMMENTS

Reviewer #1 (Airway inflammation, type 2 immunity) (Remarks to the Author):

The authors demonstrate convincingly that vaccination against IL-4 and IL-13 can be achieved in wild type mice that are tolerized to these cytokines and furthermore that such vaccination is highly effective at preventing and attenuating HDM-mediated allergic airway disease. The results furthermore indicate that neutralizing antibody levels are quite durable over periods of up to a year. The experimental design, data description and analysis, and conclusions are outstanding. The figures are also of outstanding quality.

Response: We thank Reviewer #1 for the positive comments about our work.

Major remaining questions and concerns:

1. How well can humans in general be vaccinated against these cytokines?

Response: We provide here a proof-of-concept of the efficacy of IL-4/IL-13 kinoids using mouse models, including humanized mice in which we tested a vaccine targeting the human cytokines. However, next steps of the development of this vaccine will include a toxicity study in cynomolgus monkeys, followed by an early phase trial in human to prove that this vaccine strategy can be applied in humans.

While this question can only be answered with a clinical study, it is important to note that a similar vaccination strategy has already been tested in human: a phase 2b trial recently assessed the effect of a kinoid targeting IFN- α (IFN-K) in $n=185$ adults with active systemic lupus erythematosus (SLE). This vaccine strategy has proven to be safe and to induce a neutralizing antibody response against IFN- α in 91% of treated patients (Houssiau *et al. Ann Rheum Dis*, 2020 Mar;79(3):347-355). We are thus confident that the kinoid strategy could also be efficient against IL-4 and IL-13 in humans, and are pursuing the development of this vaccine in order to start clinical trials within two years. In a follow-up study of the phase I/II trial with the IFN-K (Ducreux *et al. Rheumatology*, 2016), it was noted that most patients had a ≥ 10 -fold reduction in anti-IFN- α antibody levels one year after the first injection of the vaccine. Neutralizing antibodies were still detectable at last follow-up visit in 29% (6/21) patients who received IFN-K (range of persistence: 0.45–4.2 years). Although the strength and duration of the antibody response induced by the IL-4/13 kinoids will need to be determined in trials, they may be similar to the responses observed with the IFN- α kinoid. In our pre-clinical models, we observed 60 % persistence one year after primary immunization. We now added this information in the discussion part:

Pages 14-15 lines 303-316 “We also provided a proof-of-concept of the efficiency of kinoids targeting human IL-4 and IL-13 in mice humanized for these cytokines and their receptor IL-4R α . These promising results will now need to be confirmed in clinical studies. In this regard, while IL-4 and IL-13 vaccines have never been tested in human, a kinoid targeting interferon alpha (IFN- α) has recently been tested in a phase 2b study in 185 adults with active systemic lupus erythematosus⁴¹. This IFN- α kinoid induced a neutralizing response against IFN- α in 91% of treated patients, with an acceptable safety profile⁴¹. In a follow-up study of the phase I/II trial, it was noted that most patients had a ≥ 10 -fold reduction in anti-IFN- α antibody levels one year after the first injection of the vaccine⁴². Neutralizing antibodies were still detectable at last follow-up visit in 29 % (6/21) patients who received the IFN- α kinoid (range of persistence: 0.45-4.2 years). Although the strength and duration of the antibody response induced by the human IL-4/13 kinoids will need to be determined in clinical trials, they may be similar to the responses observed with the IFN- α kinoids. In our pre-clinical models, we observed 60 % persistence one year after primary immunization”.

2. Will the vaccination approach be effective in persons at the extremes of age-young children and the elderly?

Response: We thank Reviewer #1 for this important comment. IL-4/13 kinoids are conjugate vaccines similar to polysaccharidic vaccines in which T cell help is provided by the carrier protein. For polysaccharidic vaccines, the clear benefits of conjugate vaccines in improving the protective responses of the immature immune systems of young infants, adults and the senescent immune systems of the elderly have been made clear (Pichichero, *Hum Vaccin Immunother*, 2013, 9:12, 2505–2523; Jackson et al. *Vaccine*, 2013 Aug 2;31(35):3577-84). It is thus possible that we would observe similar effect for IL-4/13 kinoids, and in any case clinical trial will need be conducted to demonstrate immunogenicity in these populations.

3. What is the long-term effect of having no IL-4/13 and no means of readily correcting this deficiency if needed? Only human studies can of course answer these questions, but the discussion is quite brief and could be expanded to consider these issues not already addressed.

Response: We thank Reviewer #1 for this comment. Long-term targeting of IL-4 and IL-13 could in theory affect many innate and adaptive immune responses. In particular, a key role for both cytokines has been demonstrated in mouse models of host defense against helminths. Regarding risks of helminth infection in humans, more than 1.5 billion people are currently infected with helminths worldwide (According to the World Health Organization [<https://www.who.int/news-room/fact-sheets/detail/soil-transmitted-helminth-infections>]). However, to the best of our knowledge no clear data are reported on the effect of the therapeutic anti-IL-4R α mAb dupilumab on the incidence and severity of helminths infections. This is likely due, at least in part, to the fact that dupilumab is typically administered in countries with low helminth infection rates. More “long-term” safety data are now available for dupilumab in both atopic dermatitis (AD) and asthma with several large clinical studies with treatment periods from 52 to 96 weeks. Overall, these studies demonstrate a good safety profile, which argues in favor of the feasibility of long-term targeting of IL-4 and IL-13. We now added this information in the discussion part:

Pages 15-16 lines 318-333: “Besides their detrimental role in allergies, IL-4 and IL-13 also play important protective and immunoregulatory functions. In particular, these cytokines can induce host defense responses against helminths infections, and have been implicated in the promotion of anti-inflammatory and tissue repair phenotypes in macrophages⁴³⁻⁴⁵. Thus, even though we did not observe apparent side effects of the vaccines in a one-year follow-up study in mice, further work is required to evaluate whether residual IL-4 and IL-13 activity after vaccination with kinoids is sufficient to sustain protective type 2 immune responses. This question is particularly important since treatment with IL-4/IL-13 kinoids in human would induce an antibody response which would likely last several months. Although data on the long-term effects of drugs targeting IL-4, IL-13 or other type 2 cytokines in human are still limited, it is important to note that several large clinical studies are now available for the anti-IL-4R α mAb with treatment periods from 52 to 96 weeks in both asthma and atopic dermatitis^{9,46-50}. Overall, these studies demonstrate a good safety profile, which argues in favor of the feasibility of long-term targeting of IL-4 and IL-13 with a vaccine strategy. Interestingly, the most common side effect noted is conjunctivitis, although this seems to be restricted to atopic dermatitis patients, as it is not observed for patients with moderate-to-severe asthma^{9,46-48,51}”.

Other questions/issues possibly useful for considering in either the introduction or discussion:

1. Why did the authors not choose a simpler vaccination approach by choosing IL-4R α as the antigen?

Response: We decided to target IL-4 and IL-13 rather than IL-4R α for three main reasons. First, this dual vaccination strategy leaves open the possibility to target only IL-4 or only IL-13 in other diseases, shall we find that such pathology is mainly driven by one of the cytokines. Also - and most importantly - dupilumab is produced as an IgG4 which has no functional effector interactions with Fc γ R_s nor complement. This antibody format was chosen in order to reduce risks of antibody-dependent cellular toxicity (ADCC). In the case of the vaccine, a polyclonal antibody response against IL-4R α would likely induce ADCC against a range of cells expressing IL-4R α . In addition, such polyclonal response against the receptor could also be at risk of inducing antibodies which activate IL-4R α . We thus decided to rather target the soluble cytokines in order to mitigate these risks. We now discuss this at the end of the introduction part:

Page 4 lines 78-84: “Based on these partial results, and on the superior clinical efficacy in human asthma of targeting both IL-4 and IL-13 signaling (i.e. dupilumab) rather than targeting either IL-4 or IL-13 alone¹³⁻¹⁵, we hypothesized that a dual vaccination against IL-4 and IL-13 would be particularly potent at reducing the severity of chronic asthma. We decided to target IL-4 and IL-13 rather than IL-4R α to minimize risks that the vaccine would induce antibodies which could activate this receptor and/or induce antibody-dependent cellular toxicity (ADCC)”.

2. While the approach may be very helpful in asthma, it is likely to be equally or even vastly more useful in other conditions such as chronic urticaria; severe allergic rhinitis; chronic idiopathic anaphylaxis; food allergies; and chronic rhinosinusitis.

Response: We fully agree with Reviewer #1 that this vaccine approach could be useful in many conditions, not only asthma. Given the fact that dupilumab is already approved for asthma, we decided to first demonstrate the efficacy of the vaccine using validated asthma models. However, our laboratories have several studies that are now planned to further assess the efficacy of the vaccine in models of atopic dermatitis, food allergy, anaphylaxis, etc... We now acknowledge the fact that the use of this anti-IL-4/IL-13 vaccine could be extended to many other conditions in the last sentence of the discussion part:

Page 16 lines 338-340: “These results pave the way for the clinical development of an efficient long-term vaccine against asthma and other IL-4- and IL-13-mediated diseases, such as food allergies, atopic dermatitis or chronic urticaria”.

3. The approach ultimately works because it inhibits activation (by IL-4 and IL-13) of the transcription factor STAT6. Limiting the effectiveness of the approach-same concern with dupilumab-is that there are other soluble factors distinct from IL-4/13 that activate STAT6 in the allergic context, including TSLP (Han et al., J Invest Derm 2014) and cleaved fibrinogen (FCPs; Millien et al., Science 2013)

Response: We thank Reviewer #1 for highlighting the fact that other factors could drive STAT6 activation in individuals vaccinated with IL4/13 kinoids. As stated by the Reviewer, this is also the case with dupilumab, which nevertheless is efficient in a majority of treated patients. However, this could explain why we observed residual allergic inflammation, low levels of IgE, and normal IgG1 levels in mice vaccinated with IL-4/13 kinoids. We now added this information in the discussion part:

Page 14 lines 304-307: "STAT6 activation is a key step in IL-4/IL-13 signaling⁶. However, it is important to note that STAT6 can also be activated by other mediators, including TSLP³⁹ or cleaved fibrinogen⁴⁰. This could also contribute to the residual allergic inflammation, IgE and IgG1 levels observed in mice vaccinated with IL-4/IL-13 kinoids".

Other comments:

1. Line 191: humans, not men.

Response: We now corrected this mistake.

2. Line 56: though, not tough

Response: We now corrected this mistake.

Reviewer #2 (Allergy, asthma) (Remarks to the Author):

The work by Conde and colleagues reports on the effects of autovaccination for IL-4 and/or IL-13 in a murine model of allergic asthma to house dust mite. Using this elegant methodology, they were able to demonstrate the efficacy in this preclinical model that is relevant to targeted therapy of type 2 asthma, in particular with allergic phenotype, with the approved biologic dupilumab (anti-IL-4Ra mAb). Only partly overlapping were observed when comparing IL-4 and IL-13 (and dual) blockade, consistently with previous studies. The study was very well designed, using a relevant experimental model and readouts. Some issues could however be addressed to further elucidate the mechanisms of action.

Response: We wish to thank Reviewer #2 for the positive assessment of our work, and these helpful comments.

Specific comments.

1. It is quite clear from the data that downstream targets of IL-4 and/or IL-13 are effectively altered following the vaccination protocol. It should be studied whether it also affected the functionality of type 2 T cells, either Th2 cells or ILC2 as well as the production of IL-5 which is shared by those cells and mediates tissue influx and activation of eosinophils. In addition, it would be interesting to look whether it targets mainly effector eosinophils and not the recently reported regulatory eosinophils in the

lung.

Response: We agree with Reviewer #2 that these are important aspects which were not included in the first version of our manuscript. We have now quantified IL-5 in bronchoalveolar lavage (BAL) fluid collected 24h after the last challenge with HDM. Our data indicate that vaccination against IL-4 and IL-13 markedly reduces levels of IL-5 in BAL fluid, which is in full agreement with the reduced eosinophilia in vaccinated mice. These new data are now presented in **Fig 1h**, and discussed in the main text:

Page 7 lines 134-136: "Consistent with these results, levels of IL-5 - a type 2 cytokine which plays a key role in eosinophilic inflammation⁵ - were also markedly reduced in BAL fluid of HDM-treated mice vaccinated with IL-4-K and/or IL-13-K (**Fig. 1h**)."

In addition, we assessed intracellular levels of IL-4 and IL-13 in CD4⁺ T cells by flow cytometry in single cell suspensions of lung tissues collected 24h after the last HDM challenge. Our data, which are presented in **Supplementary Fig. 8a and 8b**, indicate that the vaccine does not abrogate the ability of these cells to produce IL-4 and IL-13. We also quantified the number of IL-4⁺ ILC2 (identified as CD3⁺KLRG1⁺ cells) in lung tissue by immunofluorescence. Similar to our data on CD4 T cells, we observed a slight elevation of IL-4⁺ ILC2 in the lungs of HDM-treated mice as compared to PBS controls. However, IL-4⁺ ILC2 numbers were not reduced in vaccinated mice. These new data are presented in **Supplementary Fig. 8c and 8d**. Altogether, these results suggest that vaccination against IL-4 and IL-13 does not abrogate the ability of T_H2 and ILC2 to produce these cytokines, but rather induces antibodies which neutralize these cytokines once released. We now discuss this in the main text of our manuscript:

Page 7 lines 136-142: "By contrast, we observed similar low levels of IL-4⁺ and IL-13⁺ CD4⁺ T cells (**Supplementary Fig. 8, a and b**), as well as similar numbers of IL-4⁺ type 2 innate lymphoid cells (ILC2) (**Supplementary Fig. 8, c and d**) in the lungs of HDM-treated controls and vaccinated mice. Altogether, these results suggest that vaccination with IL-4-K or IL-13-K does not abrogate IL-4 and IL-13 production in this asthma model, but rather induces antibodies which neutralize these cytokines once released".

Finally, as suggested by the Reviewer, we quantified numbers of tissue-resident eosinophils in the lungs of control mice and mice vaccinated against IL-4 and IL-13. We found no difference between both groups, indicating that the vaccine has no effect on the number of regulatory eosinophils in the lung at baseline. We discussed with Dr. Thomas Marichal, corresponding author of the paper describing these regulatory eosinophils (Mesnil *et al. J Clin Invest* 2016), who validated our gating strategy (we now thank him in the acknowledgement section). These data are presented in **Supplementary Fig. 9b and 9c**, and discussed in the main text of our manuscript:

Page 7 lines 147-150: "In addition, while numbers of effector eosinophils were markedly reduced in the lungs of vaccinated mice (**Fig. 1, f and g**), we observed no effect of the vaccine on the number of tissue-resident regulatory eosinophils²³ (**Supplementary Fig. 9, b and c**)".

In addition, we found that the vaccine mainly targets effector eosinophils and, to a lesser extent, B cells, but that numbers of neutrophils, macrophages and T cells were

similar in the BAL fluid of controls and vaccinated HDM-treated mice. These data are presented in a novel **Supplementary Fig. 10**, and discussed in the main text of our manuscript:

Page 7 lines 150-153: “The effects of the vaccine on leukocytes appeared to be mostly restricted to effector eosinophils and to a lesser extent B cells, as we observed similar levels of neutrophils, macrophages and T cells in the BAL fluid of HDM-treated mice vaccinated with IL-4-K and IL-13-K or CRM₁₉₇ as a control (**Supplementary Fig. 10**)”.

2. The authors documented by invasive lung function tests that the protocol affected AHR, at least in the prophylactic setting. therefore, in addition to mucus production, it should be confirmed whether the treatment approach (in particular targeting IL-13 or both IL-4/IL-13) did alter smooth muscle hypertrophy that is classically seen (at least in human asthma, but also in chronic experimental asthma).

Response: We thank Reviewer #2 for this suggestion. We now stained alpha smooth muscle actin (α -SMA) in lung tissue of all groups of mice collected 24h after the last challenge with HDM, in order to quantify smooth muscle hypertrophy in our model. Although the staining of α -SMA gave a clear peribronchial signal, we did not observe significant differences between PBS controls and HDM-treated mice (please see **Fig. A**, below). Thus, unfortunately the asthma model we used, although leading to clear AHR, eosinophilia and mucus production, does not permit us to assess the effect of the vaccine on smooth muscle hypertrophy.

Fig A. Effect of HDM-treatment and vaccination with IL-4/IL-13 kinoids on alpha-smooth muscle actin in the lung. Mice vaccinated with IL-4-K and/or IL-13-K (or CRM₁₉₇ as a control) were sensitized and challenged intranasally with HDM as described in Fig. 1a. **a.** Representative lung sections stained with anti-alpha smooth muscle actin (α -SMA) antibodies (which stains the peri-bronchial smooth muscle cell layer, red) and DAPI (which stains nucleus, blue) 24 h after the last challenge. **b.** Quantification of the thickness of the peribronchial α -SMA⁺ smooth muscle cell layer. Results show mean values from individual bronchi (8-12 bronchi were analyzed from a total of 4 mice/group). Bars indicate means \pm SEMs.

3. It would be extremely interesting to see whether this strategy is applicable in human asthma patients, although the reviewer acknowledge this will be the next step through an early phase trial. It could be however important, in order to get some translational applicability, to discuss the contrast between the efficacy of IL-13 vaccination in mice

and the failure of anti-IL-13 mAbs in human (severe) asthma. Similarly, it should be discussed to what extent findings could be extrapolated to type 2 non allergic asthma (which may be also very eosinophilic, as some cases associated with nasal polyposis), notably because dupilumab efficacy does not rely on the allergic nature of asthma. Could the Authors comment on those translational aspects?

Response: Thank you for giving us the opportunity to discuss the translational applicability of the vaccine. We think that the contrast between the beneficial effect of the anti-IL-13 vaccine in mice and the failure of anti-IL-13 treatments in clinical trials is likely to be due to the difference in terms of outcomes analyzed. In fact, LAVOLTA and STRATOS, two large clinical trials, failed to demonstrate an effect of lebrikizumab and tralokinumab (two anti-IL-13 mAbs) on the reduction of asthma exacerbation (Hanania *et al. Lancet Respir Med*, 2016; Panettieri Jr *et al. Lancet Respir Med*, 2018). The mechanisms of asthma exacerbation are not fully understood, and are likely to rely on a combination of complex parameters, including eosinophilic (and/or neutrophilic) airway inflammation, IgE effector mechanisms and airflow obstruction. In our mouse study, some of these features were reduced upon vaccination against IL-13 (e.g. AHR and mucus production), while some others required combined vaccination against IL-4 and IL-13 (e.g. IgE production and lung eosinophilia).

In agreement with the effect of IL-13 vaccination on AHR in our mouse study, an increase in FEV1 (forced expiratory volume in 1 second) has been observed in clinical trials conducted with anti-IL-13 mAbs, especially when focusing on patients with biomarker evidence of type 2 asthma (high blood eosinophil counts or periostin concentrations, which is induced by IL-4 and IL-13) (Piper *et al. Eur Respir J*, 2013; Brightling *et al. Lancet Respir Med*, 2015 ; Hanania *et al. Lancet Respir Med*, 2016).

We speculate that combined blockade of IL-4 and IL-13 pathways is probably also required in order to efficiently reduce most features of asthma in human, leading to reduced asthma exacerbation. This would provide a potential explanation for the superior clinical efficiency of dupilumab over various therapeutic anti-IL-4 or IL-13 mAbs in asthma. As highlighted by the Reviewer, dupilumab has a beneficial effect on severe eosinophilic asthma regardless of the allergic status in humans. It is thus tempting to speculate that the IL-4/IL-13 vaccine could also have beneficial effects in type 2 non-allergic asthma (but of course this will need to be proven through clinical trials).

We now discuss these points in the discussion part:

Page 13 lines 268-281 "In line with our data in mice, IL-13 also plays an important role in controlling airway reactivity in human asthma. Indeed, an increase in FEV1 (forced expiratory volume in 1 second) has been observed in clinical trials conducted with anti-IL-13 mAbs^{14,32,33}, especially when focusing on patients with biomarker evidence of type 2 asthma (e.g., high blood eosinophil counts or periostin concentrations, which is induced by IL-4 and IL-13³⁴). However, LAVOLTA and STRATOS, two large clinical trials, failed to demonstrate an effect of lebrikizumab and tralokinumab (two anti-IL-13 mAbs) on asthma exacerbation^{14,35}. Thus, we speculate that combined blockade of IL-4 and IL-13 pathways is probably also required in order to efficiently reduce most features of asthma in human, as we observed here in mouse models with the IL-4 and IL-13 vaccines. This provides a potential explanation for the superior clinical efficiency

of dupilumab (which blocks both IL-4 and IL-13 signaling) over various therapeutic anti-IL-4 or IL-13 mAbs in asthma¹³⁻¹⁵, and prompted us to focus on the dual vaccine for further evaluation in a therapeutic protocol”.

Reviewer #3 (Asthma, allergy) (Remarks to the Author):

The manuscript by Conde et al. describes the effects of a dual vaccination against IL-4 and IL-13 in a mouse model of allergic asthma. As worded by the authors, the rationale for this work is that “dupilumab - a monoclonal antibody (mAb) against IL-4Ra that blocks both IL-4 and IL-13 signaling - is efficient at decreasing the rate of severe exacerbations, and at improving lung function in patients with moderate-to-severe asthma. However, use of this (or any other) mAb in chronic asthma is limited by high cost and the need to perform injections over years to lifelong.” This is a somewhat biased view. Costs are currently high but there is no reason they could not be lowered in the future.

Response: We thank Reviewer #3 for these comments. We certainly hope that the price of dupilumab (and other mAbs) will be lowered in the future. However, there is no sign of such trend at this time. Indeed, dupilumab is protected by patents until October 2029, which clearly blocks development of similar mAbs, as exemplified by the recent decision of the European Patent Office to invalidate Immunex European patent claiming mAbs that target IL-4R (http://sitepilot.firmseek.com/client/mckool2/www/media/news/159_2019%2002%2014%2096%20Final%20Written%20Decision.pdf). In addition, discounts for originator biologics facing biosimilar competition remain low (below 20%) compared to generic, and will not happen before all patents expire.

Current cost per year is around \$37,000 for dupilumab. So even if prices were to be lowered in the future, this certainly would still represent substantial costs linked to manufacturing costs and repeated injections, as compared to a vaccine approach. Finally, the emergence of additional strategies to treat allergic diseases, such as a vaccine approach, could help lowering prices of therapeutic mAbs, including dupilumab, in the future.

Most importantly, though, the need for lifelong therapy has the positive flipside that it makes it possible to stop treatment as needed – an option that does not appear to be available in the authors' approach. Type 2 immunity (both innate and adaptive) has protective, tissue repair-inducing properties, not to mention its protective effects against parasites and possibly viruses such as SARS-CoV-2. In this context, long-lasting elimination or even just suppression of type 2 responses by vaccination is a questionable, concerning goal. A more nuanced and practical alternative to addressing the asthma problem may be to prevent its inception by engaging natural regulatory/balancing mechanisms.

Response: We fully agree with the Reviewer that a very important point for future clinical development of this vaccine would be its safety with regards to the ever-growing list of potential protective type 2 immune responses. This can include tissue repair, metabolism, neuroinflammation, cancer, helminth infection, etc. We fully

acknowledge the importance and concern of targeting such protective pathways, and this will be one of the main focus of our work in future years. As suggested by the Editor, and Reviewers #1 and #2, we have now added an extensive discussion on the implication, caveats, and potential long-term adverse effects in human application, as well as a deeper characterization of the effect of the vaccine on key innate cells including ILC2, neutrophils and macrophages, as well as T_H2 cells and the newly-described lung-resident regulatory eosinophils. In addition, we found that IgG antibody responses are intact in vaccinated mice, which is important in the current vision that COVID pandemic may last several years, or may be a recurrent epidemic. All these points are detailed in our responses to your specific comments, below, and in the response to the specific comment 1 from Reviewer #2 (pages 6-8, above).

Some specific issues:

1. The statement that "type 2 inflammation characterized by high levels of cytokines such as interleukin-4 (IL-4) and IL-13, high levels of IgE antibodies, and airway eosinophilia occurs in approximately 50 % of patients with asthma" is overall true but incomplete. Except for high IgE, these responses are essentially tissue-specific, not systemic. Therefore, it is not clear that a systemic suppressive approach would be optimal.

Response: We thank Reviewer #3 for this comment. We have now revised our sentence to reflect the fact that IL-4 and IL-13 responses and eosinophilia are mostly tissue-specific.

Page 3 lines 51-54: "type 2 inflammation characterized by production of interleukin-4 (IL-4) and IL-13 in the lung, airway eosinophilia and high levels of IgE antibodies occurs in approximately 50 % of patients with asthma^{1,5}".

We agree that it would be very interesting in the future to develop a strategy to selectively block IL-4 and IL-13 locally in the airways. However, while local delivery is efficient for small molecules, and is currently under investigation for antibodies (especially small antibody fragments such as Fab fragments, nanobodies or ScFv), such approach is to date not applicable for a vaccine strategy. Interestingly, an anti-IL-13 Fab fragment has been tested in a model of allergic asthma in cynomolgus monkeys (Lightwood *et al. Am J Respir Crit Care Med*, 2018). The Fab fragment had moderate effects on BAL eosinophils and markedly reduced BAL IL-5, which is in agreement with our data in mice obtained with the IL-13 kinoid. However, even though the Fab fragment was delivered by nebulization, significant levels of the antibody fragment could be detected in the blood. Therefore, even with this approach, systemic effects of the anti-IL-13 therapy cannot be totally ruled out. We now discuss this in the discussion part of the manuscript:

Pages 13-14 lines 277-286: "The extent to which the protective effects of anti-IL-4 and IL-13 therapy in asthma reflects local blockade of the cytokines in the lung vs. systemic effects is still not fully understood. Interestingly, local delivery of an anti-IL-13 Fab fragment by nebulization has been tested in a model of allergic asthma in cynomolgus monkeys³⁶. This Fab had moderate effects on BAL eosinophilia but markedly reduced BAL IL-5 levels, which is in full agreement with our data obtained in HDM-exposed mice vaccinated with the IL-13 kinoid. However, even though the anti-IL-13 Fab was

delivered by nebulization, significant levels of the antibody fragment could be detected in circulation. By extension one may consider that systemic effects of the anti-IL-13 kinoid therapy are conceivable, even if we did not detect any significant systemic effects”.

2. Given the topic of this work, the authors may want to discuss the ligand/receptor interaction patterns that underpin the differential asthma-promoting effects of IL-4 and IL-13 despite their shared receptor.

Response: We thank Reviewer #3 for this suggestion. We now added a discussion of the ligand/receptor interaction patterns in the discussion part of our manuscript:

Pages 12-13 lines 254-261: “Such non-overlapping functions could be explained, at least in part, by the different receptor requirement for the two cytokines. The type 1 IL-4 receptor only recognizes IL-4, and is a heterodimer of IL-4R α and the common gamma chain (γ c)^{6,29,30}. The type 2 IL-4 receptor binds both IL-4 and IL-13, and is a heterodimer of IL-4R α and IL-13R α 1^{6,29,30}. In addition, IL-13 (but not IL-4) also binds IL-13R α 2, which was long thought to function as a decoy receptor that limits the activity of IL-13^{6,29,30}. However, evidence indicates that IL-13 signaling through IL-13R α 2 can induce production of TGF- β 1³¹”.

3. As mentioned above, reading that "Such neutralizing capacity could still be detected in more than 60 % of the mice over one year after primary immunization" is perplexing, given the life span of a mouse – especially because there is no switch off mechanism for this approach.

Response: While the lifespan of a laboratory mouse is around two years, we think that the duration of an immune response in mice cannot be extrapolated to human only based on the lifespan expectancy. In addition, in our mouse models we used 4 (in WT mice) or 5 (in humanized mice) injections of IL-4/13 kinoids in a relatively short period of time. Both the doses and timing of injections in human will need to be defined based on a dose-range-finding study in monkeys followed by clinical trials.

A kinoid directed against IFN- α (IFN-K) has already been tested in patients with systemic lupus erythematosus (SLE). A phase 2b trial recently conducted in $n=185$ adults showed that the vaccine strategy is safe and induces a neutralizing antibody response against IFN- α in 91% of treated patients (Houssiau *et al. Ann Rheum Dis*, 2020 Mar;79(3):347-355). In a follow-up study of the phase I/II trial with the IFN-K (Ducreux *et al. Rheumatology*, 2016), it was noted that most patients had a ≥ 10 -fold reduction in anti-IFN- α antibody levels one year after the first injection of the vaccine. Neutralizing antibodies were still detectable at last follow-up visit in 29% (6/21) patients who received IFN-K (range of persistence: 0.45–4.2 years). Although the strength and duration of the antibody response induced by the IL-4/13 kinoids will need to be determined in clinical trials, they may be similar to the responses observed with the IFN- α kinoid (i.e. only be transient in the absence of booster doses of the vaccine). We now added this information in the discussion part:

Pages 14-15 Lines 303-316: “We also provided a proof-of-concept of the efficiency of kinoids targeting human IL-4 and IL-13 in mice humanized for these cytokines and their

receptor IL-4R α . These promising results will now need to be confirmed in clinical studies. In this regard, while IL-4 and IL-13 vaccines have never been tested in human, a kinoid targeting interferon alpha (IFN- α) has recently been tested in a phase 2b study in 185 adults with active systemic lupus erythematosus⁴¹. This IFN- α kinoid induced a neutralizing response against IFN- α in 91% of treated patients, with an acceptable safety profile⁴¹. In a follow-up study of the phase I/II trial, it was noted that most patients had a ≥ 10 -fold reduction in anti-IFN- α antibody levels one year after the first injection of the vaccine⁴². Neutralizing antibodies were still detectable at last follow-up visit in 29 % (6/21) patients who received the IFN- α kinoid (range of persistence: 0.45-4.2 years). Although the strength and duration of the antibody response induced by the human IL-4/13 kinoids will need to be determined in clinical trials, they may be similar to the responses observed with the IFN- α kinoids. In our pre-clinical models, we observed 60 % persistence one year after primary immunization”.

4. For the reasons repeatedly mentioned above, the proposed vaccination strategy might be translationally significant (perhaps!) only as a treatment, not as prevention. However, the effects of dual vaccination in the therapeutic model are tenuous, especially for AHR.

Response: We agree that this vaccine should first be considered as a treatment, not as a prevention strategy. This is the reason why we also chose to test the vaccine using a therapeutic protocol in mice. We agree that the effect of the vaccine on AHR in this model is somehow less evident than in the prophylactic protocol, however both airway resistance and elastance were significantly reduced by the vaccine. Importantly, in this long model, mice were exposed to high doses of HDM twice a week for a total of 18 weeks, which permitted us to initiate the injection of the vaccine once significant AHR and eosinophilia had developed (as shown in **Supplementary Fig 14**). However, the repeated HDM injections over such a long period of time probably also initiated resolution pathways, as the overall AHR (in HDM-treated non-vaccinated mice) was lower in this model than in the shorter prophylactic protocol. This might, at least in part, account for the less evident effect of the vaccine on AHR in the therapeutic vs. prophylactic protocol. The therapeutic effects of the vaccine were however more evident when analyzing levels of IgE, airway eosinophilia and mucus production, which overall demonstrates the efficiency of the vaccine in this setting.

5. The authors should directly investigate, not just discuss, how their vaccination strategy affects protective type 2 responses, both innate and adaptive.

Response: As stated by the Reviewer, targeted therapies against IL-4/IL-13 could affect many innate and adaptive immune responses, and assessing the safety/potential long-term effects of the vaccine is our main priority prior to any clinical development of such vaccine. While we are planning to assess the effect of the vaccine in various models of innate immune responses, and in host defense against parasites, these models require new ethic protocol approval, and will require substantial time to assess the effects of the vaccine in an unbiased manner (assessing effects at different time-points after vaccination; comparing results to that obtained with IL-4R α KO mice, or IL-4/13 KO mice; comparing responses of hIL-4/hIL-13^{KI}; hIL-4R α ^{KI} humanized mice vaccinated with hIL-4/hIL-13 kinoids vs. treatment with dupilumab).

Regarding risks of helminth infection, more than 1.5 billion people are currently infected with helminths worldwide (According to the World Health Organization [<https://www.who.int/news-room/fact-sheets/detail/soil-transmitted-helminth-infections>]). However, to the best of our knowledge no clear data are reported on the effect of dupilumab therapy (or other drugs targeting type 2 cytokines) on the incidence and severity of helminths infections. This is likely due, at least in part, to the fact that dupilumab is most often administered in countries with low helminth infection rates. More “long-term” safety data are now available for dupilumab in both atopic dermatitis (AD) and asthma with several large clinical studies with a 52-week treatment period in both asthma and atopic dermatitis (AD). Overall, these studies demonstrate a good safety profile, which argues in favor of the feasibility of long-term targeting of IL-4 and IL-13. We now added this information in the discussion part:

Pages 15-16 lines 318-333: “Besides their detrimental role in allergies, IL-4 and IL-13 also play important protective and immunoregulatory functions. In particular, these cytokines can induce host defense responses against helminths infections, and have been implicated in the promotion of anti-inflammatory and tissue repair phenotypes in macrophages⁴³⁻⁴⁵. Thus, even though we did not observe apparent side effects of the vaccines in a one-year follow-up study in mice, further work is required to evaluate whether residual IL-4 and IL-13 activity after vaccination with kinoids is sufficient to sustain protective type 2 immune responses. This question is particularly important since treatment with IL-4/IL-13 kinoids in human would induce an antibody response which would likely last several months. Although data on the long-term effects of drugs targeting IL-4, IL-13 or other type 2 cytokines in human are still limited, it is important to note that several large clinical studies are now available for the anti-IL-4R α mAb with treatment periods from 52 to 96 weeks in both asthma and atopic dermatitis^{9,46-50}. Overall, these studies demonstrate a good safety profile, which argues in favor of the feasibility of long-term targeting of IL-4 and IL-13 with a vaccine strategy. Interestingly, the most common side effect noted is conjunctivitis, although this seems to be restricted to atopic dermatitis patients, as it is not observed for patients with moderate-to-severe asthma^{9,46-48,51}”.

6. The dissection of the differential effects of IL-4 and IL-13 in asthma pathogenesis, while interesting, is not novel.

Response: We are glad that the Reviewer considers that our dissection of the differential effects of IL-4 and IL-13 through separate vaccination is interesting. We agree that such differential effects are not novel, and we cite and discuss previous findings in both human and mouse models on this topic. However, we consider that it was essential to show that such results can be reproduced through a vaccine strategy, and to demonstrate that optimal protective effects of the vaccine can only be obtained upon dual targeting of IL-4 and IL-13.

7. The humanized mice used in these experiments need to be described in some detail. In particular, it is important to specify whether they are knock-ins (as their denomination acronym might indicate) or they still carry the endogenous mouse genes. Result interpretation will be deeply affected in the latter case.

Response: We thank Reviewer #3 for this very important comment. We generated hIL-4/hIL-13^{KI}; hIL-4R α ^{KI} mice by syntenic replacement of two mouse loci: that

encoding *Il4/Il13* and the second encoding *Il4ra*, with the corresponding human segment of DNA. Therefore, these hIL-4/hIL-13^{KI}; hIL-4R α ^{KI} mice express the human genes in the place of the mouse gene, and thus cytokine receptor interactions in these animals model those of the human proteins. We now added a scheme (new **Supplementary Fig. 17**) describing the strategy chosen to replace mouse IL-4/IL-13 and IL-4R α by the human proteins in these mice. We now further characterized this novel humanized mouse strain, and results of these experiments are presented in a novel **Figure 4**. In particular, we confirmed by ELISA that splenocytes from these mice (but not from WT mice) can release human IL-4. We obtained similar results with human IL-13, and further showed that significant levels of human IL-13 can be detected in BAL fluid from hIL-4/hIL-13^{KI}; hIL-4R α ^{KI} mice following chronic intranasal sensitization and challenge.

Page 10 lines 206-225: “Low interspecies similarity of IL-4 (~44 %) and IL-13 (~55 %) between mice and human would render mouse IL-4-K and IL-13-K highly immunogenic in humans, and less potent to generate neutralizing responses. We therefore developed and characterized kinoids eliciting an immune response against human IL-4 and IL-13 (hIL-4-K and hIL-13-K) (**Supplementary Fig. 16**), and used mice humanized for IL-4, IL-13 and for their common receptor chain IL-4R α (hIL-4^{KI}; hIL-13^{KI}; hIL-4R α ^{KI} mice) by syntenic replacement of two mouse loci: that encoding *Il4/Il13* and the second encoding *Il4ra*, with the corresponding human segment of DNA. These hIL-4/hIL-13^{KI}; hIL-4R α ^{KI} mice express the human genes in the place of the mouse gene, and thus cytokine receptor interactions in these animals model those of the human proteins (**Supplementary Fig. 17**). We confirmed that splenocytes from hIL-4/hIL-13^{KI}; hIL-4R α ^{KI} mice, but not from WT mice, release human IL-4 upon stimulation with PMA and ionomycin (**Fig. 4a**). We obtained similar results with human IL-13, and further showed that significant levels of human IL-13 can also be detected in BAL fluid from hIL-4/hIL-13^{KI}; hIL-4R α ^{KI} mice following chronic intranasal sensitization and challenge (**Fig. 4, b and c**). We further confirmed expression of human IL-4Ra (and lack of expression of mouse IL-4Ra) by immunohistology in skin samples from hIL-4^{KI}; hIL-13^{KI}; hIL-4R α ^{KI} mice (**Fig. 4, d and e**). Finally, we showed that intranasal challenge with recombinant human IL-4 or human IL-13 leads to eosinophilia, lung inflammation and mucus production in hIL-4^{KI}; hIL-13^{KI}; hIL-4R α ^{KI} mice (**Fig. 4, f-j**). Altogether, these data demonstrate that this new hIL-4^{KI}; hIL-13^{KI}; hIL-4R α ^{KI} humanized mouse strain both produces and responds to hIL-4 and hIL-13”.

REVIEWERS' COMMENTS

Reviewer #1 (Remarks to the Author):

All concerns adequately addressed; no further concerns.

Reviewer #2 (Remarks to the Author):

The Authors have appropriately addressed my concerns/remarks.

Reviewer #3 (Remarks to the Author):

Thank you for the opportunity to read the revised version of Dr. Reber's manuscript describing the effects of dual vaccination against IL-4 and IL-13 in chronic allergic experimental asthma models. As I stated in my initial review, the experiments are overall well done and well described, and the latest data provide further mechanistic details. I remain unimpressed by the results obtained using the vaccine as a treatment, but this is a relatively minor concern at this stage. My major concern is that I still cannot agree with the rationale and premise of this work – not because I think this vaccine may fail to suppress Th2 responses, but because it may succeed. The authors have not convinced me that disabling for a long, undefined period of time an entire arm of the immune system - one that likely evolved NOT to make us allergic or asthmatic but to protect the integrity of our mucosal tissues - is a sound, valid strategy against asthma. In fact, the Authors' arguments in this respect are rather lame (allergic asthma and active systemic lupus erythematosus are in distinct leagues of severity, and parasitic infections are only one of several conditions raising concern).

I realize that decisions about manuscripts ultimately rest with the Editors, whom we reviewers can only advise, and I am tempted to "hide" by providing a purely technical review of methodologies and figures, all of which are of good quality. The problem is, this work is framed explicitly as a preclinical study, and its publication would provide a steppingstone towards clinical trials. Therefore, I cannot in good conscience endorse it.

REVIEWERS' COMMENTS

Reviewer #1 (Remarks to the Author):

All concerns adequately addressed; no further concerns.

Reviewer #2 (Remarks to the Author):

The Authors have appropriately addressed my concerns/remarks.

Reviewer #3 (Remarks to the Author):

Thank you for the opportunity to read the revised version of Dr. Reber's manuscript describing the effects of dual vaccination against IL-4 and IL-13 in chronic allergic experimental asthma models. As I stated in my initial review, the experiments are overall well done and well described, and the latest data provide further mechanistic details. I remain unimpressed by the results obtained using the vaccine as a treatment, but this is a relatively minor concern at this stage. My major concern is that I still cannot agree with the rationale and premise of this work – not because I think this vaccine may fail to suppress Th2 responses, but because it may succeed. The authors have not convinced me that disabling for a long, undefined period of time an entire arm of the immune system - one that likely evolved NOT to make us allergic or asthmatic but to protect the integrity of our mucosal tissues - is a sound, valid strategy against asthma. In fact, the Authors' arguments in this respect are rather lame (allergic asthma and active systemic lupus erythematosus are in distinct leagues of severity, and parasitic infections are only one of several conditions raising concern).

I realize that decisions about manuscripts ultimately rest with the Editors, whom we reviewers can only advise, and I am tempted to "hide" by providing a purely technical review of methodologies and figures, all of which are of good quality. The problem is, this work is framed explicitly as a preclinical study, and its publication would provide a steppingstone towards clinical trials. Therefore, I cannot in good conscience endorse it.

Response: We thank Reviewer #3 for the thoughtful review of our manuscript, and for stating that our experiments are overall well done and well described. We do hear the argument of the Reviewer that targeting IL-4 and IL-13 might on the long term reduce protective type 2 immune responses. The goal of this first pre-clinical work was to provide a proof-of-concept of the efficacy of this vaccine approach in a model of asthma in mice.

In line with the concerns raised by this Reviewer, we discuss the potential long-term effects of targeting IL-4/IL-13 in response to parasite infections and beyond, and the need for further work to define these potential long-term effects, as well as to better define the duration of the immune response induced by the vaccine. In an effort to present a balanced/unbiased discussion, we also discuss the overall good safety profile of the anti-IL-4R mAb dupilumab in asthma patients, even in some studies where continuous treatment was given for up to 96 weeks. No major adverse effects were reported and a clear health and quality of life benefit resulted from this long treatment period. We cannot speculate on the duration of anti-IL-4 and anti-IL-13 vaccination in humans for now, but immune responses to other conjugate vaccines wane with time and require boosters to maintain efficacy. Further studies are required to evaluate the duration of our proposed therapeutic vaccine regimen, but it may not

deprive patients from these TH2 cytokines more than dupilumab long treatments have done so far, which did not reveal major susceptibility to other diseases or infections.

While type 2 immune responses undoubtedly have protective effects, it is also clear that more and more people develop aberrant type 2 responses that can lead to severe chronic diseases, and that need to be reduced on the long term.